# Balanced Conic Rectified Flow

**Shin seong Kim**
Yonsei University
tltydl2@yonsei.ac.kr

**Mingi Kwon**
Yonsei University
kwonmingi@yonsei.ac.kr

**Jaeseok Jeong**
Yonsei University
jete_jeong@yonsei.ac.kr

**Youngjung Uh**[*]
Yonsei University
yj.uh@yonsei.ac.kr

## Abstract

Rectified flow is a generative model that learns smooth transport mappings between two distributions through an ordinary differential equation (ODE). The model learns a straight ODE by reflow steps which iteratively update the supervisory flow. It allows for a relatively simple and efficient generation of high-quality images. However, rectified flow still faces several challenges. 1) The reflow process is slow because it requires a large number of generated pairs to model the target distribution. 2) It is well known that the use of suboptimal fake samples in reflow can lead to performance degradation of the learned flow model. This issue is further exacerbated by error accumulation across reflow steps and model collapse in denoising autoencoder models caused by self-consuming training. In this work, we go one step further and empirically demonstrate that the reflow process causes the learned model to drift away from the target distribution, which in turn leads to a growing discrepancy in reconstruction error between fake and real images. We reveal the drift problem and design a new reflow step, namely the *conic reflow*. It supervises the model by the inversions of real data points through the previously learned model and its interpolation with random initial points. Our conic reflow leads to multiple advantages. 1) It keeps the ODE paths toward real samples, evaluated by reconstruction. 2) We use only a small number of generated samples instead of large generated samples, 600K and 4M, respectively. 3) The learned model generates images with higher quality evaluated by FID, IS, and Recall. 4) The learned flow is more straight than others, evaluated by curvature. We achieve much lower FID in both one-step and full-step generation in CIFAR-10. The conic reflow generalizes to various datasets such as LSUN Bedroom and ImageNet. The project page is available at https://grainsack.github.io/BC_rectified_flow_project_page/.

## 1 Introduction

Rectified flow [9, 28, 24, 25, 23] demonstrates state-of-the-art image generation with fewer sampling steps than diffusion models [6, 43, 37, 15, 41]. $k$-rectified flow involves $k$ reflow steps that make ODE paths smooth and straight [27]. This allows the model to generate high-quality images simply and efficiently in just one or a few steps. Intriguingly, *all rectified flow models* (Flux, SD3, and AuraFlow) achieve state-of-the-art quality with *1-rectified flow* and require about 30 NFEs (number of function evaluations) [9].

---

[*]Corresponding author.

39th Conference on Neural Information Processing Systems (NeurIPS 2025).

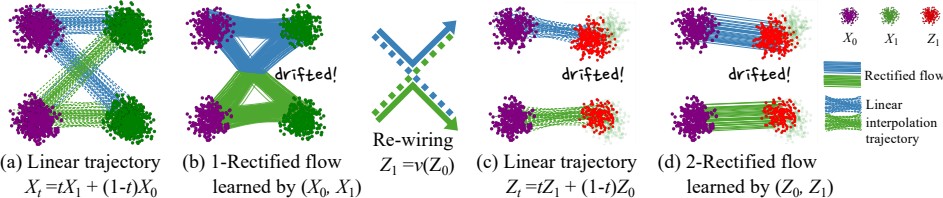

Figure 1: **Problem of rectified flow.** (a) By randomly pairing data $X_0 \sim \pi_0$ and $X_1 \sim \pi_1$, linear interpolation trajectories are defined. (b) The 1-rectified flow connects $X_0$ and $X_1$ with a learned velocity field which is potentially inaccurate. After the 1-rectified flow, the trajectories are rewired to avoid crossing. (c) The trajectories for reflow are defined as linear interpolation trajectories between $Z_0$ and the generated $Z_1 = v(Z_0)$. Note that $Z_1$ is drifted away from $\pi_1$. (d) Consequently, the 2-rectified flow has a velocity field drifted away from $X_1$.

In this paper, we identify key limitations of $k$-rectified flow stemming from its reflow procedure, and propose to use real images and their inversions, combined with a Slerp-based perturbation loss, to address them. Our core insight is that the degradation in performance cannot be fully explained by error accumulation or vanishing weight norms alone [18, 56]. Instead, we observe that the reflow process itself induces a drift away from the target distribution, leading to a measurable discrepancy between the model's behavior on real and fake samples, which we later analyze through reconstruction error. 1) **The flow drifts away from the real distribution.** For example, in training a 2-rectified flow, random noise vectors and their generated images from the 1-rectified flow are reused as supervisory targets, which causes the ODE paths to diverge from real data. We empirically show that this results in a reconstruction performance gap between real and fake samples, and that Slerp-based supervision for real samples helps maintain alignment with the true distribution. 2) **The number of fake samples required for reflow is prohibitively large.** Our method leverages real samples and their conic neighbors to guide the flow with far fewer generated pairs, while maintaining competitive or superior performance. This efficiency makes the method scalable and less dependent on synthetic supervision. 3) **The reflow process degrades image quality in full-step generation.** By providing trajectory-level supervision rooted in real data geometry, our method enhances generation quality across 1-step, few-step, and full-step regimes, and mitigates the degradation in standard reflow training.

As a result, we successfully demonstrate better performance than existing $k$-rectified flow models. On CIFAR-10, we reduce the FID (Fréchet Inception Distance, [14]) of the existing 2-rectified flow from 12.21 to 5.98 while using only 7.2% of the generative pair, and show that the curvature of the ODE paths became straighter. We also introduce a new method for calculating curvature, which explains the time distribution sampling method. We provide various ablation studies and show that even with simple fine-tuning, the performance of existing $k$-rectified flow can be significantly improved.

## 2 Rectified Flow

Rectified flow [27] is a generative model that solves an ordinary differential equation (ODE) to induce a transition trajectory between two given data distributions $\pi_0$ and $\pi_1$. Data $X_0 \sim \pi_0$ and $X_1 \sim \pi_1$ define linear trajectories $X_t = (1 - t)X_0 + tX_1$ for $t \in [0, 1]$, as illustrated in Figure 1a. Then, a rectified flow $v$ is an ODE on time $t$ parameterized by $\theta$:

$$\frac{dZ_t}{dt} = v_\theta(Z_t, t) := \frac{1}{t}(Z_t - \mathbb{E}[(X_1 - X_0)|X_t = Z_t]) \tag{1}$$

We omit $\theta$ for brevity. Liu et al. [27] propose a simplified mean squared error (MSE) loss for an ODE neural network to train velocity field $v : \mathbb{R}^n \to \mathbb{R}^n$ as follows:

$$\arg\min_\theta \mathbb{E}\left[\left\|X_1 - X_0 - v(tX_1 + (1-t)X_0, t)\right\|^2\right] \tag{2}$$

With $t \sim \text{Uniform}([0, 1])$. In image generation tasks, $X_0 \sim \pi_0$ and $X_1 \sim \pi_1$ are random noise and real images, sampled from a Gaussian distribution and the data distribution, respectively. Once the model has learned the velocity field, the rectified flow rewires the trajectories in a non-crossing manner due to the inherent properties of ODEs, which enforce uniqueness and smoothness in

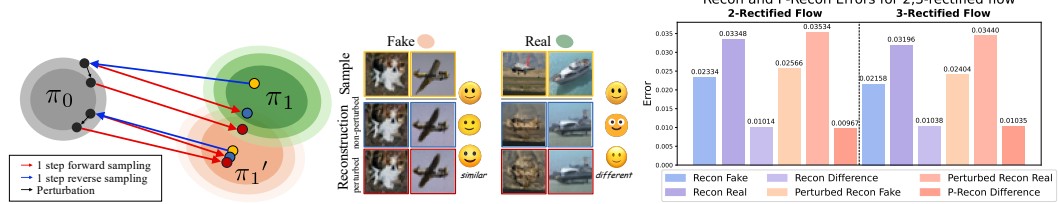

Figure 2: **(a) 2-rectified flow overfits fake samples.** Following the reverse and forward 2-rectified flow, fake images inherently return at similar images with or without perturbation at $\pi_0$. In contrast, real images return at different images and it is worse with perturbation, implying overfitting. **(b)** Reconstruction discrepancies emerge between real and fake images due to the use of fake-only pairs.

trajectory evolution, preventing paths from intersecting in the continuous-time dynamics, as depicted in Figure 1b. It constitutes a 1-rectified flow model denoted by $\mathbf{Z} = \texttt{RectFlow}((X_0, X_1))$.

The $k$-rectified flow model learns a straighter sampling trajectory by repeating *reflow procedure* k times as follows. Following $\mathbf{Z}^k$ from $Z_0^k$ induces a generated pair $(Z_0^k, Z_1^k)$ where $(Z_0^0, Z_1^0) = (X_0, X_1)$. It redefines the linear interpolation trajectory $\mathbf{Z}_t^{k+1} = (1-t)Z_0^k + tZ_1^k$ for $t \in [0, 1]$, as shown in Figure 1c. Then, fine-tuning a velocity field $v$ using Eq. (2) with $(Z_0^k, Z_1^k)$ instead of $(X_0, X_1)$ constitutes $\mathbf{Z}^{k+1} = \texttt{RectFlow}((Z_0^k, Z_1^k))$.

According to optimal transport theory, [45, 44, 10, 11] coupling the generated pairs $(Z_0, v(Z_0))$ ensures that the interpolation trajectory preserves the marginal distributions of the original and target domains, as well as the linear interpolation trajectory between them [21, 2]. $k$-rectified flow has superior quality of few-step sampling by straighter sampling trajectory, as shown in Figure 1d.

## 3 Improved Techniques for Reflow Step

In this section, we discuss the generated fake pairs in the original rectified flow and their problems. Then, we introduce real pairs and balanced-conic reflow which directly supervise the flow to reach the real data distribution. Finally, we provide detailed training configurations.

### 3.1 Reflow steps drift the flow away from the real distribution.

As shown by Liu et al. [27], the trajectory $Z_t^k$ between the generated pairs in the reflow process becomes smoother and straighter with each iteration of reflow, because the ODE induces a deterministic smooth solution while preserving the same marginal distribution as the original trajectory. This straightened path is essential for generating high-quality images with a small number of sampling steps rather than SDE-based generative models [15, 37, 35]. We use subscript $F$ to denote the **fake pairs** from the original rectified flow as follows:

$$(Z_{0,F}^k, Z_{1,F}^k) := (Z_0^k, Z_1^k) \tag{3}$$

Where $Z_0^k \sim \pi_0$ and $Z_1^k = v(Z_0^k)$.[2] To simplify notation and avoid confusion, we will denote the fake pair as $(Z_{0,F}, Z_{1,F})$ when we do not need to consider the reflow step and, denote the $k$-th order of the rectified flow as $(Z_{0,F}^k, Z_{1,F}^k)$.

Beyond known issues such as error accumulation and model collapse [18, 56], we provide empirical evidence that reflow steps cause a distributional drift, observed through growing reconstruction discrepancies between real and fake samples. We provide empirical evidence of the accumulating drifts and suboptimality of the learned rectified flow. Figure 2a illustrates faithfulness and continuity[3] of a 2-rectified flow on both fake samples and real samples. As expected, fake images are mostly reconstructed by inversion and generation following the 2-rectified flow, i.e., $Z_1 \simeq v(v^{-1}(Z_1))$. Also, an inversion of a fake sample and its perturbation land at similar images, i.e., $v(v^{-1}(Z_1)) \simeq$

---

[2]For brevity, we denote the forward generation process at the t-th sampling step as $X_0 + \int_0^1 v_t(X_t, \cdot)\, dt := v(X_0)$ and backward process as $X_1 + \int_0^1 v_t^{-1}(X_t, \cdot)\, dt := v^{-1}(X_1)$, where $v^{-1} = -v$.

[3]We use the notion of continuity as in Lipschitz continuity: the generated images should be similar with the similar latents.

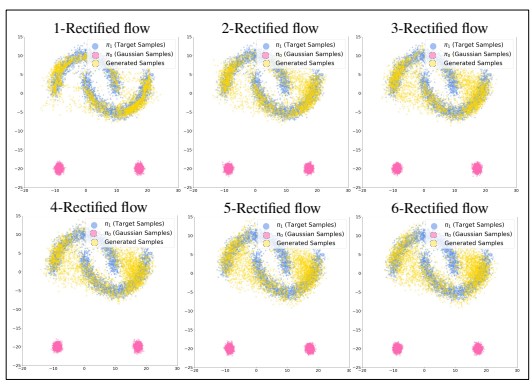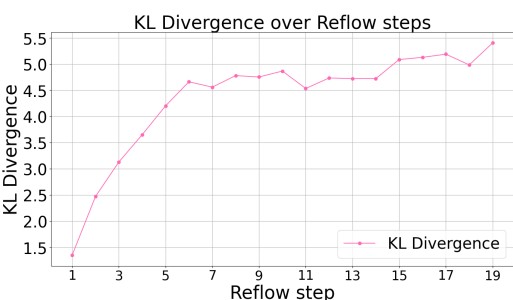

Figure 3: **(a)** As reflow steps increase, generated samples diverge from the target distribution. **(b)** This drift is further evidenced by the rising KL divergence from the real data distribution.

$v(v^{-1}(Z_1) + \varepsilon z)$ where we perturb it with $\varepsilon = 0.05$. In contrast, real images lose the main object by inversion and generation following the 2-rectified flow, i.e., $X_1 \neq v(v^{-1}(X_1))$. Furthermore, real images are vulnerable to perturbation on their inversion, i.e., $v(v^{-1}(X_1)) \neq v(v^{-1}(X_1) + \varepsilon z)$. To evaluate the faithfulness of a rectified flow to a dataset $X$, we measure the error between the sample and its reconstruction [4, 33] via the flow:

$$L_2^{\text{recon}}(X) = \mathbb{E}_{x \sim X}\left[\|x - v(v^{-1}(x))\|_2\right] \tag{4}$$

Instead of Lipschitz continuity, we practically evaluate the continuity of a rectified flow near samples from a dataset $X$ by measuring a perturbed reconstruction error:

$$L_2^{\text{p-recon}}(X, \varepsilon) = \mathbb{E}_{x \sim X, z \sim \pi_0}\|x - v(v^{-1}(x) + \varepsilon z)\|_2, \tag{5}$$

where $\varepsilon$ is the strength of perturbation. The lower perturbed reconstruction error near the real samples $L_2^{\text{p-recon}}(X_1)$ indicates the more continuous generative model near the real samples.[4] Figure 2b compares $L_2^{\text{recon}}$ of real and fake samples, and $L_2^{\text{p-recon}}$ near real and fake samples. $L_2^{\text{recon}}$ is higher at the real samples than the fake samples. It indicates that the 2,3-rectified flow drifts away from the real samples. Furthermore, $L_2^{\text{p-recon}}$ is lower near the fake samples than the real samples. It indicates that the 2-rectified flow suffers from crossing between real samples.

Critically, the original reflow accumulates the drift over recursive reflow steps. This drift is an innate phenomenon because the supervision from a shifted distribution does not steer the flow toward the real distribution. Figure 3a provides empirical evidence of the accumulating drift in a toy `two moons` dataset [32]. The successive reflow steps cause the fake (yellow) data to diverge further from the real (blue) target distribution. Furthermore, Figure 3b illustrates the progressive increase in KL divergence between the fake distribution and the target distribution, providing clear evidence of this phenomenon.[5] We provide a solution to mitigate this issue in Section 3.3 and 3.4.

## 3.2 Real pair

The previous subsection has unveiled the pitfall of supervision using fake pairs: samples from the domain distribution, e.g., Gaussian, and their codomain following a rectified flow. Instead of the fake pairs, we propose to use the real samples and their inverse following a reverse rectified flow, defined by :

$$\text{Real pair}(Z_{0,R}, X_1) := (v^{-1}(X_1), X_1) \tag{6}$$

with $X_1 \sim \pi_1$ where $v$ is the 1-rectified flow and we abuse the term *real pair* although the $Z_{0,R}$ is not real. As in the original rectified flow, where it was optionally provided, it is safe and easy to use reverse rectified flow without stochasticity because it inherently produces a deterministic solution, and using real images does not contradict the original purpose because the noise $v^{-1}(X_1)$ is generated using $v^{-1}$. To avoid confusion, from now on, we will refer to $(X_0, v(X_0))$ as a *fake pair* (generated pair), where $v(X_0)$ is a fake (generated) image, and $(v^{-1}(X_1), X_1)$ as a *real pair*, where $X_1$ is a real image.

---

[4]We measure the reconstruction and perturbed reconstruction error with 1-step Euler sampling.

[5]The KL divergence is measured between Gaussian mixture approximations of the fake and real samples.

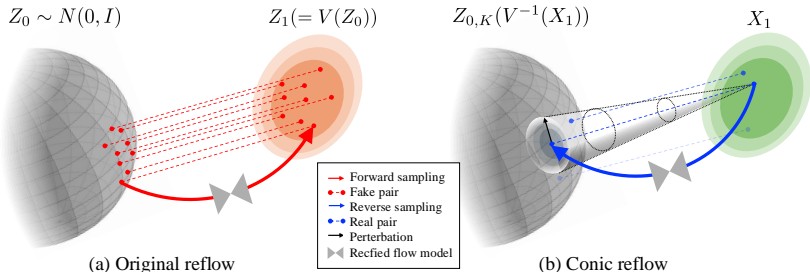

$Z_0 \sim N(0, I)$   $Z_1(= V(Z_0))$   $Z_{0,K}(V^{-1}(X_1))$   $X_1$

Forward sampling
Fake pair
Reverse sampling
Real pair
Perterbation
Recfied flow model

(a) Original reflow          (b) Conic reflow

Figure 4: **Illustration of original fake pairs and our real pairs.** (a) The original rectified flow supervises 2-rectified flow with fake pairs $(Z_0, v_\theta(Z_0))$. (b) Our conic reflow supervises 2-rectified flow with real pairs $(v_\theta^{-1}(X_1), X_1)$ and their conic neighbors.

## 3.3 Conic reflow

Building upon the basic pairing of real samples with their reverse noises, we introduce conic reflow, which expands their influence on the domain distribution to their neighboring areas, as shown in Figure 4b. When we train the model, we use spherical linear interpolation (Slerp) between the reverse noise $Z_{0,R}$ and a randomly sampled noise $\epsilon \sim \mathcal{N}(0, I)$ with the interpolation ratio $\zeta$:

$$\text{Slerp}(Z_{0,R}, \epsilon, \zeta) = \frac{\sin((1 - \zeta)\phi)}{\sin(\phi)} Z_{0,R} + \frac{\sin(\zeta\phi)}{\sin(\phi)} \epsilon, \tag{7}$$

where $\phi = \arccos(Z_{0,R} \cdot \epsilon)$ denotes the angle between $Z_{0,R}$ and $\epsilon$. Then we define a conic inverse from a real sample $X_1$:

$$\text{Conic}(X_1, \epsilon, \zeta, t) = tX_1 + (1 - t)\text{slerp}(Z_{0,R}, \epsilon, \zeta), \tag{8}$$

where $Z_{0,R} = v_\theta^{-1}(X_1)$, and $t \in [0, 1]$. During training, we sample $\epsilon$ and $\zeta$ multiple times to let the target flow stochastically cover the nearby domain. As the collection of the paths over multiple iterations looks like a cone, we name our method as *conic reflow*. The schedule of interpolation weight $\zeta$ is deferred to Section 3.5. Our training objective with conic reflow is:

$$\hat{\theta} = \arg\min_\theta \int_0^1 \mathbb{E}\Big[w_t \big\| X_1 - \text{slerp}(Z_{0,R}, \epsilon, \zeta) - v_\theta\big(\text{Conic}(X_1, \epsilon, \zeta, t)\big)\big\|^2\Big] dt \tag{9}$$

Where $t \sim \exp([0, 1])$, $\zeta \sim \text{slerp schedule}([0, 1])$, $\epsilon \sim \mathcal{N}(0, I)$, and $w_t$ is weighting function (default=1). Slerp is commonly used for interpolation in the noise space of generative models, as it preserves vector magnitudes on the Gaussian hypersphere and enables smooth semantic transitions [47, 16]. In our case, Slerp serves as a geometry-aware regularizer that improves the alignment between numerically inverted real samples and the underlying data manifold. Rather than simply reusing generated samples, we supervise the model using real samples and their perturbed inversions to reduce reconstruction loss discrepancies observed during reflow. This approach not only mitigates distributional drift but is also aligned with adversarial robustness strategies studied in inverse problems and reconstruction-based neural networks [3, 1]. From this perspective conic reflow provides localized supervision that encourages stability and continuity around real data points in the noise space.

## 3.4 Balanced conic rectified flow

We design a new reflow procedure which consists of our conic reflow and the original reflow. For each training iteration, we design different training schemes for real pairs and fake pairs. We alternate between conic reflow steps with real pairs $(Z_{0,R}, X_1)$ and original reflow steps with fake pairs. It encourages the trajectories to head toward the exact real distribution while fake pairs ensure the entire domain distribution to receive supervision. Fake Pairs: For fake pairs, we proceed with the reflow process exactly as it was done in the original rectified flow model. Its training objective $\hat{\theta}$ as follows:

$$\arg\min_\theta \mathbb{E}\left[\|Z_{1,F} - Z_{0,F} - v_\theta(tZ_{1,F} + (1 - t)Z_{0,F})\|^2\right], \tag{10}$$

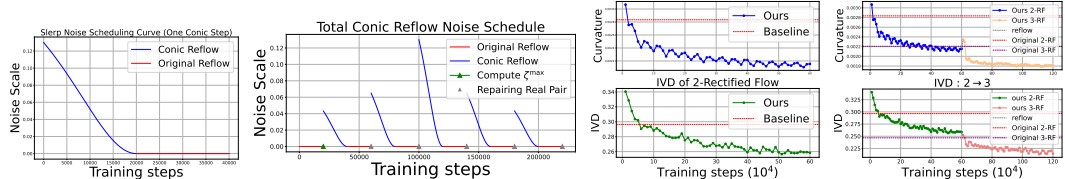

Figure 5: **(a)** Slerp noise schedule for single conic reflow and **(b)** Total slerp schedule. **(c)** Curvature and IVD during training, and **(d)** From 2- to 3-rectified flow.

with $t \sim \exp([0,1])$. The entire training objective of our method is as follows for the given fake pair$(Z_{0,F}, Z_{1,F})$ and real pair $(Z_{0,R}, X_1)$:

$$
\min_{v} \int_0^1 \left[ \left\| \chi_{\text{fake}} \cdot \left( \dot{Z}_{t,F} - v_\theta(Z_{t,F}) \right) + \chi_{\text{real}} \cdot \left( X_1 - \text{slerp}(Z_{0,R}, \epsilon, \zeta) - v_\theta\big(\text{Conic}(X_1, \epsilon, \zeta, t)\big) \right) \right\|^2 \right] dt
\tag{11}
$$

where $\chi_{\text{fake}}$ and $\chi_{\text{real}}$ are indicator functions for given index subsets $U_{\text{real}}$ and $U_{\text{fake}}$ such that $U_{\text{real}} \cup U_{\text{fake}} = \mathbb{N}$. Then, for $i \in \mathbb{N}$:

$$
\chi_{\text{fake}} = \begin{cases} 1 & \text{if } i \in U_{\text{fake}} \\ 0 & \text{else} \end{cases}, \qquad \chi_{\text{real}} = \begin{cases} 1 & \text{if } i \in U_{\text{real}} \\ 0 & \text{else}. \end{cases}
\tag{12}
$$

The notation $\zeta$, Conic$(\cdot)$, $\epsilon$, and $w_t$ follows the definitions in Section 3.3.

## 3.5 Detailed training schemes

In this section, we provide a more detailed explanation of our proposed training schemes, including visualizations of the Slerp scheduling and maximum Slerp noise magnitude. The maximum magnitude of the Slerp noise, $\zeta_{\max} \in (0, 0.5]$ is selected in accordance with our intuition that it is the point where the discrepancy between the perturbed reconstruction errors of real and fake samples is maximized. After a warm-up phase of training steps, we compute $\zeta_{\max}$ using 10,000 images each from the real and fake. The value is selected based on the following hyperparameter $\zeta^{\max}$:

$$
\zeta^{\max} := \max_{\zeta \in (0, 0.5]} \; \mathbb{E}_{\substack{x \sim X_1 \\ z \sim Z_{1,F}}} \left[ \left\| v_\theta\left( \text{Slerp}(z_{0,R}, \epsilon, \zeta) \right) - x \right\|_2 - \left\| v_\theta\left( \text{Slerp}(z_{0,F}, \epsilon, \zeta) \right) - z \right\|_2 \right]
\tag{13}
$$

where $\epsilon \sim \mathcal{N}(0, I)$, with $z_{0,R} = v_\theta^{-1}(x)$ and $z_{0,F} = v_\theta^{-1}(z_{1,F})$ denoting the inverse noise estimates for the real and fake samples, respectively. Figure 5(a) shows a single conic reflow noise schedule in training steps (For CIFAR-10 and ImageNet, we set $\zeta^{\max}$ to 0.13 and 0.23, respectively). Each conic is trained to progressively reduce the noise scale over time. Specifically, the Slerp noise schedule $\zeta(t')$ is defined as $\zeta(t') := \zeta^{\max} \cdot \frac{2t'^2}{1+t'^2}$, $t' \in [0, 1]$, where $t' = 1$ corresponds to the start of training and $t' = 0$ to the end. This design follows the intuition from traditional diffusion models [15], where noise is progressively reduced to guide samples toward realism. To strengthen the real image trajectory during training, we periodically update the real sample pairs used in the reflow process. Figure 5(b) illustrates an example of conic reflow with updated real pairs, assuming 220K total training steps for visualization. When the number of updates is $2K$, we schedule the maximum noise magnitude $\zeta^{\max}$ by scaling it according to the pattern $[K, K-1, \ldots, 1, 2, \ldots, K]$, assigning $\zeta^{\max}/K$ to the smallest value and $\zeta^{\max}$ to the midpoint. The noise increases linearly in the first half and decreases symmetrically in the second half. The full pseudocode for our training method is provided in Appendix K.

# 4 Experiments

We conducted experiments to evaluate the effectiveness of our method. Our findings demonstrate: Superiority over original reflow in terms of (1) Quality of the results, (2) Straightness of the flow, (3) Mitigation of distribution shift, as well as (4) Ablation study, (5) Generalization to other datasets.

| Method | NFE (↓) | IS (↑) | FID (↓) |
|---|---|---|---|
| **One-Step Generation (Euler solver, N=1)** | | | |
| 1-Rectified Flow | 1 | 1.13 (9.08) | 378 (6.18) |
| *2-Rectified Flow* | | | |
| Original (+*Distill*) | 1 | 8.08 (9.01) | 12.21 (4.85) |
| **Ours** (+*Distill*) | 1 | **8.79 (9.11)** | **5.98 (4.16)** |
| Rf++† [24] | 1 | 8.87 | 4.43 |
| Rf++† (+**ours**) | 1 | 8.87 | **4.22** |
| *3-Rectified Flow* | | | |
| Original (+*Distill*) | 1 | 8.47 (8.79) | 8.15 (5.21) |
| **Ours** (+*Distill*) | 1 | **8.84 (8.96)** | **5.48 (4.68)** |
| **Full Simulation (Runge–Kutta (RK45), Adaptive N)** | | | |
| 1-Rectified Flow | 127 | **9.60** | **2.58** |
| *2-Rectified Flow* | | | |
| Original | 110 | 9.24 | 3.36 |
| **Ours** | 104 | **9.30** | **3.24** |
| *3-Rectified Flow* | | | |
| Original | 104 | 9.01 | 3.96 |
| **Ours** | 98 | **9.14** | **3.70** |

Table 1: One-step and full-simulation comparison of 2,3 Rectified Flows on CIFAR-10.

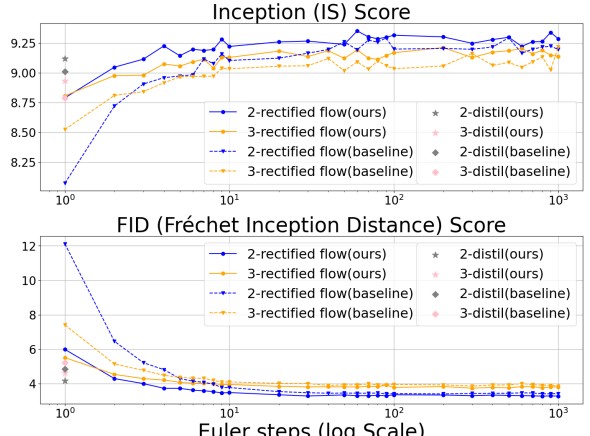

Figure 6: CIFAR-10 generation quality across Euler steps.

**Experimental setup**  Most of our experiments are conducted on CIFAR-10 [20]. The IVD, curvature, reconstruction, and perturbed ($0.05\varepsilon$, 1-step) reconstruction error values reported were computed using 10,000 random samples, with the expectation taken over these samples. We employ Scipy's RK45[46], a 5(4) Runge-Kutta method with adaptive step size and step count determined by specified tolerances, following the same parameters [37]. Further details on the training configurations are provided in Appendix J.

## 4.1 Image quality

Our method achieves better FID and IS scores across all sampling steps, i.e., 1-step, few-step, and full-step generations, as shown in Table 1 and Figure 6. Notably, we use only 300K fake pairs, significantly fewer than the 4M fake pairs used in the original rectified flow, demonstrating the efficiency of our approach. Furthermore, our method outperforms the original in generation quality with RK sampling, even with a lower NFE. We also apply our method to Rectified++ [24] and compare generation quality. While RF++† uses 800K synthetic pairs, our variant uses 600K fake pairs and 50K real images, achieving better FID. This indicates that our method generalizes well to reflow-based generative models. Detailed settings for RF++† are provided in Appendix J.

Additionally, our method produces images with superior quality even when applying the same distillation in the original rectified flow [27] as compared to solid (ours) and dashed (original) lines, and star (ours) and rectangle (original) in Figure 6. These results suggest that our method produces a more favorable initial velocity field than the original rectified flow. A detailed comparison of precision and recall scores is provided in Appendix B. Furthermore, we show in Appendix D that fine-tuning the original rectified flow using only real images and their reverse noise pairs effectively reduces reconstruction discrepancy and quickly improves generation quality. Additional performance comparisons on more complex datasets and higher-resolution images are presented in Sections 4.5 and 4.6. To further contextualize the performance of reflow-based methods, we also report the unconditional generation quality of pretrained diffusion models on CIFAR-10 in Appendix N.

## 4.2 Straightness

We evaluate trajectory straightness using curvature, following prior works [27, 23]. Straighter trajectories reduce discretization error under few step solvers, improving sample quality [39, 5]. For any continuously differentiable process $\mathbf{Z} = \{Z_t\}$ , the curvature is measured by :

$$S(\mathbf{Z}) = \int_0^1 \mathbb{E}\left[\left\|(Z_1 - Z_0) - \dot{Z}_t\right\|^2\right] dt \tag{14}$$

Additionally, it is known that $\mathbf{S(Z)} = 0$ indicates exact straightness.

**Relationship between curvature and initial velocity delta (IVD)**  While curvature captures trajectory straightness, it does not fully explain 1-step sampling quality, which depends solely on the initial velocity. We propose Initial Velocity Delta (IVD) to directly evaluate the accuracy of the

initial velocity and its impact on 1-step generation. The calculation method for IVD is provided in the equation below:

$$IVD(\mathbf{Z}, t_0) = \mathbb{E}\left[\left\|(Z_1 - Z_0) - \dot{Z}_{t_0}\right\|^2\right] \tag{15}$$

**Curvature and initial velocity delta.**  Our approach demonstrates improved trajectory straightening and better preservation of the initial velocity direction. As shown in Figure 5(c), our method consistently achieves lower curvature and IVD values than the original, indicating a more stable trajectory even with fewer fake pairs. Furthermore, Figure 5(d) highlights that applying an additional reflow step from 2-rectified to 3-rectified flow further reduces both curvature and IVD, reinforcing the effectiveness of our training method.

## 4.3    Reconstruction with perturbation to address drift from the real distribution

We empirically show that incorporating real data pairs via Slerp-based supervision in the reflow step significantly improves the preservation of real image trajectories. As illustrated in Figure 7(a), the original rectified flow exhibits a clear reconstruction error gap between real and fake images, whereas our method progressively narrows this gap during training. Figure 7(b) highlights two key aspects. First, in terms of reconstruction error, our method better preserves real trajectories and mitigates overfitting to fake samples. Second, the lower perturbed reconstruction error suggests that the model more accurately aligns the velocity field around real images, leading to improved robustness against perturbations.

## 4.4    Ablation study

We report results from an ablation study on various settings of our proposed framework, comparing four configurations: (1) without Slerp noise; (2) reflow using just a single real pair; (3) our full method; and (4) the original 2-rectified flow. The comparison focuses on 1-step generation quality, curvature, IVD(Initial Velocity Delta), reconstruction error, and perturbed reconstruction error.

### 4.4.1    Benefits of incorporating real data via slerp-based perturbation training

| Model | FID | IS | Curvature | IVD | Recon Real | Recon Fake | Perturbed Real | Perturbed Fake |
|---|---|---|---|---|---|---|---|---|
| Original | 12.21 | 8.08 | 0.002837 | 0.295078 | 0.033668 | 0.024106 | 0.035481 | 0.026270 |
| **Ours* (Slerp + conic)** | **5.98** | **8.79** | **0.002295** | 0.253334 | **0.019404** | 0.023139 | **0.022382** | 0.025206 |
| Fixed Real Pair | 6.69 | 8.59 | 0.002313 | 0.242444 | 0.020227 | 0.020607 | 0.022890 | 0.022914 |
| No Slerp | 6.60 | 8.57 | 0.002322 | **0.240884** | 0.023380 | **0.020154** | 0.026063 | **0.022496** |

Table 2: An ablation table comparing various 2-rectified models across multiple metrics: FID, IS (1-step), Curvature, IVD, and errors (Recon Real/Fake, Perturbed Real/Fake).

**Benefits of adding noise via Slerp**    Adding noise via Slerp avoids trajectory crossover, preserving a straighter path relative to real images. This leads to improved trajectory quality and enhances 1-step sampling efficiency. Moreover, using Slerp results in lower reconstruction error for real images compared to not using it. This demonstrates that Slerp-based noise perturbation helps stably maintain the trajectory and local neighborhood structure of reverse noise corresponding to real images. In particular, it plays a key role in reducing the reconstruction discrepancy we observe between real and fake samples during reflow, thereby effectively mitigating the distributional drift that occurs in later iterations.

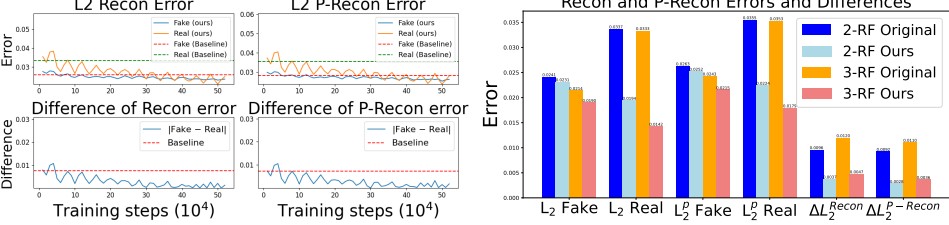

Figure 7: Reconstruction and perturbed reconstruction error across training iterations.

**Effect of real pair injection and refreshing**  Incorporating real pairs into training improves trajectory fidelity and 1-step generation quality by anchoring learning closer to the target distribution. Notably, fixing the real pair only once leads to inferior performance, while periodically refreshing reverse noise achieves better FID, IS, and lower reconstruction errors. This highlights the importance of continuously updating real-sample guidance throughout training to maintain alignment with real data trajectories.

Additional experiments, including $k = 4$ generation quality, precision/recall, low fake pair settings, and real-only supervision, are provided in Appendices C and E.

### 4.4.2 Slerp noise patterns and lerp

In this section, we show that our Slerp-based reflow method consistently improves 1-step generation quality, even when the Slerp noise pattern varies. To support this claim, we train 2-rectified flow models using different Slerp schedules, including patterns where the noise gradually increases, gradually decreases, or first increases and then decreases. We also include a baseline where Slerp is replaced with Lerp for comparison. All experiments are conducted with a batch size of 256, and training is performed for 300K iterations. Each setting uses 300K fake pairs and 60K real pairs for training on Cifar 10. Other configurations remain identical to those described in Section 4 and Appendix J.

| Category | Method | Schedule Type | IS ($\uparrow$) | FID ($\downarrow$) |
|----------|--------|---------------|-----------------|--------------------|
| **Slerp** | Ours | $\zeta_k^{\max}$: $\zeta^{\max}/K \to \zeta^{\max} \to \zeta^{\max}/K$ | **8.72** | **6.63** |
| | Strictly Increasing | $\zeta_k^{\max}$: $\zeta^{\max}/K \to \zeta^{\max}$ | 8.48 | 6.64 |
| | Strictly Decreasing | $\zeta_k^{\max}$: $\zeta^{\max} \to \zeta^{\max}/K$ | 8.45 | 6.70 |
| **Lerp** | Linear Interpolation | $(1-\zeta) \cdot v^{-1}(X_1) + \zeta \cdot \epsilon, \zeta \in [0, \zeta^{\max}]$ | 8.46 | 7.50 |

Table 3: Each method uses Slerp or Lerp-based noise trajectories. The best and second-best values are bolded and underlined, respectively.

As shown in Table 3, we observe three key findings. First, Slerp-based schedules consistently outperform Lerp in both FID and IS, suggesting that Slerp more effectively preserves the reverse noise trajectories of real images. Second, the strictly increasing noise schedule yields better performance than the decreasing one, indicating that progressive noise injection improves training stability. Third, our method achieves the highest IS while maintaining comparable FID to the best baseline, demonstrating improved sample diversity without compromising generation quality.

### 4.5 More complex dataset (Unconditional generation on ImageNet 64×64)

On ImageNet 64×64 [7], our method consistently improves unconditional generation quality over the original model. Using the same setup as CIFAR-10, we train a 1-rectified flow for 700K steps with batch size 256. The original uses **1M** fake pairs, while our method uses **600K** fake and 60K real pairs. As shown in Table 4, Although 60K real images may be insufficient to fully cover the target distribution of ImageNet compared to CIFAR-10, our method still yields substantial gains in FID and Recall. This suggests that even limited real data can provide strong guidance in mitigating distributional drift. Moreover, it improves reconstruction error, perturbed reconstruction error, recall, and precision (see Table 5 and Appendix F), demonstrating its ability to reduce distributional drift even on large-scale datasets.

### 4.6 High-resolution image generation

In this section, we assess the generalizability of our method on the LSUN Bedroom dataset [52] at a resolution of 256×256. We use the same hyperparameters, time schedule, and EMA settings as in the experiments by Liu et al. [27]. We use 60K fake pairs and 5K real pairs, while the original uses 120K fake pairs. Despite GPU limitations on the larger LSUN dataset, our method consistently outperformed the original in image quality. Figure 8 shows superior few step generation quality, and with adaptive step sampling (RK45), our approach achieved comparable quality with significantly fewer fake pairs than the original. Further details on the training configurations are provided in Appendix J. For comparisons under different random seeds, refer to Appendix I.

| Metric | FID | | IS | |
|---|---|---|---|---|
| ODE Solver | RK-45 | Euler | RK-45 | Euler |
| 1-rectified flow | 23.1 | 369.8 | 12 | 1.1 |
| 2-rectified flow | 31.2 | 39.7 | 10.8 | **10.4** |
| **Ours** | **28.2** | **37.8** | **11.4** | 10.3 |

Table 4: FID and IS across rectified flows using RK-45 and Euler solver on ImageNet.

| Model | Solver | Precision | Recall |
|---|---|---|---|
| Original | Euler 1-step | **0.4129** | 0.4604 |
| Ours | Euler 1-step | 0.3812 | **0.5325** |
| Original | RK-45 | **0.4717** | 0.5264 |
| Ours | RK-45 | 0.4432 | **0.6032** |

Table 5: Precision and recall on ImageNet.

| Solver | FID (Original/Ours) | Precision | Recall |
|---|---|---|---|
| 1-step Euler | 139.98 / **26.54** | 0.0290 / **0.4822** | 0.0220 / **0.2274** |
| RK | 24.76 / **24.14** | 0.4525 / **0.4703** | 0.2386 / **0.2388** |
| **IVD, Recon, P-Recon error (Fake/Real)** | | | |
| | IVD | $L_2^{recon}$ | $L_2^{p-recon}$ |
| Original | 1.1790 | 0.0822 / 0.1147 | 0.0820 / 0.1146 |
| Ours | **0.9103** | **0.0487 / 0.0405** | **0.0486 / 0.0407** |
| | $\Delta L_2^{recon}$ | $\Delta L_2^{p-recon}$ | |
| Original | 0.0325 | 0.0326 | |
| Ours | **0.0083** | **0.0079** | |

Euler 1-step     Euler 2-step     RK - 45

Figure 8: Visual and quantitative comparison on LSUN. **Left**: 2-row layout showing original (**top**) and ours (**bottom**) for each solver (1-step, 2-step, RK). **Right**: evaluation metrics.

# 5 Related work

Recent efforts to improve rectified flow models focus on modifying the time distribution, loss functions, or model architectures [24, 18]. Some works incorporate real data pairs or introduce discriminator-based regularization to reduce the effect of out-of-distribution samples [18]. Others analyze model degradation through the lens of denoising autoencoders, attributing performance drop to the vanishing of weight norms caused by repeated training on synthetic pairs [56]. PerFlow takes a different approach by constructing piecewise linear intermediate paths to stabilize reflow training [50]. In contrast to these works, our method introduces a perturbation-based supervision using real data inversions.

# 6 Conclusion and future work

We propose balanced conic reflow, a simple yet effective method that addresses key limitations of traditional rectified flow by incorporating real pairs through a Slerp-based perturbation strategy. Our approach improves generation quality across multiple settings, while requiring significantly fewer synthetic samples. Its plug-and-play nature makes it compatible with various flow-based generative models, such as InstaFlow and SD3 [28, 9]. Future directions include extending our method with additional loss functions or integrating it with diffusion-based synthetic supervision [24, 18]. Moreover, our strategy may complement recent frameworks such as Rectified Diffusion [48], which demonstrate that reflow-like improvements are possible even without retraining $v$-prediction-based flow matching models, where conic reflow can further help preserve real image trajectories and reduce reconstruction discrepancy without relying on linear interpolation paths or discriminator-based supervision.

# Acknowledgment

This work was partly supported by the National Research Foundation of Korea (RS-2023-00223062) and an IITP grant (RS-2020-II201361, RS-2024-00439762) funded by the Korean government (MSIT).

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
