# OpenReview forum: "Balanced Conic Rectified Flow"
_NeurIPS.cc/2025/Conference — NeurIPS 2025 poster_

### Official Review · Reviewer_srv3 · 2025-06-04

**Clarity:** 2
**Significance:** 3
**Originality:** 3
**Rating:** 4
**Confidence:** 4

**Summary:**

The authors propose conic rectified flow (RF), a new alternative to RF to mitigate the distribution drift in the successive reflow steps. The main idea of the conic RF is to design a map that transports a neighbourhood of points (rather than a single Gaussian point) to a corresponding image point $X_1$.
The experimenst show that conic RF is able to reduce the reconstruction error gap between real and fake images, and also achieves comeptetive results in real examples.

**Questions:**

## Major comments
**Definition of $v$ is not clear:**
- Line 72: ''coupling the generated pairs $(Z_0, v(Z_0))$
ensures that the interpolation trajectory preserves the marginal distributions of the original and target
domains'': Assuming $v$ is the velocity field,  I think $(Z_0, v(Z_0))$ does not preserve the marginals of the source and target distribution. In fact, $v(Z_0)$ is just the velocity field (the time argument is missing). I think it should be $(Z_0, T(Z_0))$, where $T(Z_0):= Z_0 + \int_{0}^1 v(Z_t, t)\; dt$. Do the authors mean to write $T$ instead of $v$?
- Line 87: Footnote 1 suggest that the $t$-th sampling step is denoted by $v(X_0)$ which shoudl be $X_t + v(X_t, t)dt$ (Not $X_0 + v(X_t, t)dt$) as per my understadning. Is it denoting something else?  Also, $v(Z_0^k)$ seems to denote the last sampling step instead of any $t$th sampling step. The notations are hard to understand.

- Shouldn't there be an integral over $\epsilon, \zeta$ in Eq(9)?


**What exactly is better in conic RF than regular RF?:**

- Section 3.1 argues that RF suffers from distribution drift, i.e., $T(T^{-1}(X_1)) \ne X_1$. How is this getting resolved in the Conic RF? As far as I understand, conic RF is trying to map rather an expanded neighbourhood around $T^{-1}(X_1)$ to a single point $X_1$. But, how does it stop the distribution shift?

- Prior works have shown that real data augmentation reduces model collapse. Conic RF also uses real data during training, along with fake data. It might be possible that real data augmentation is the main factor that is preventing the distribution drift, rather than the explicit design of the flow mechanism. For example, [2] shows that real data augmentation mitigates model collapse. In light of this fact, it seems that using the methodology in [2] is more practical than opting for conic RF, as conic RF is comparatively harder to train. What is the extra advantage of Conic RF? I think a discussion on this is important. Maybe adding a comparison between the method in [2] and conic RF will help.


## Minor comments

- Line 67 writes k-rectified flow, whereas Line 74 writes $k$-rectified flow. It is advisable to choose one consistent notation.

- The set $\mathbb{N}$ in Line 160 is no defined. Also, where does the index $i$ in Eq (12) occur in Eq (11)? Are you referring to the empirical version of the loss in Eq. (11)?

[1] Liu, X., Gong, C., & Liu, Q. (2022). Flow straight and fast: Learning to generate and transfer data with rectified flow. arXiv preprint arXiv:2209.03003.

[2] Zhu, H., Wang, F., Ding, T., Qu, Q., & Zhu, Z. (2024). Analyzing and Improving Model Collapse in Rectified Flow Models. arXiv preprint arXiv:2412.08175.

**Ethical Concerns:**

["NO or VERY MINOR ethics concerns only"]

**Final Justification:**

My concerns have been addressed properly.

**Limitations:**

yes

**Quality:**

3

**Strengths And Weaknesses:**

## Strength
The idea seems novel and quite effective in practice. To generate reliable results in practice, it is essential to address and resolve the issue of distribution drift in the RF reflow steps. This paper attempts to provide a solution in that direction. The experiments presented in the paper are comprehensive and showcase the usefulness of conic RF.

## Weakness
Although the idea is novel, the exact benefit of the current approach over regular RF is not clear. For example, how exactly conic RF prevents the distribution drift is not clear to me immediately. All in all, I think a stronger case needs to be made in favor of conic RF to argue the significance of conic RF over other methods (e.g., RF with real data augmentation introduced in [1] )
 Also, there are some presentation issues that I have pointed out later.


[1] Zhu, H., Wang, F., Ding, T., Qu, Q., & Zhu, Z. (2024). Analyzing and Improving Model Collapse in Rectified Flow Models. arXiv preprint arXiv:2412.08175.

---

> ### Author Rebuttal · Authors · 2025-07-31
>
> Thank you for recognizing the effectiveness and novelty of our approach, as well as for appreciating the comprehensiveness of our experiments and the practical relevance of addressing distribution drift in ReFlow.
>
> ## W1,2
>
> Thank you for this insightful comment. You are absolutely right: in this context, $v(Z_0)$ is not meant to represent the velocity field evaluated at a single point, but rather the outcome of solving the ODE defined by the velocity field. In other words, our intention was to represent the trajectory mapping $T(Z_0) := \int_0^1 v(z_t, t) dt$, and we used the notation $v(Z_0)$ as a shorthand for this operator.
>
> To simplify notation and avoid introducing an additional operator like $T$, we opted for this form. However, as you point out, this may lead to confusion, especially when integration is not explicitly stated.
>
> We also acknowledge that in Footnote 1, the integral form was unintentionally omitted, which may have compounded this ambiguity. We will revise the footnote to explicitly include the integral definition, and we will consider introducing auxiliary notation (e.g., $T(Z_0)$) in the revised version to improve clarity and prevent confusion for future readers.
>
> We sincerely appreciate your careful reading and helpful suggestion.
>
> ## W3
>
> Thank you for pointing this out. Indeed, $\epsilon$ and $\zeta$ are sampled during training and introduce stochasticity. However, in Equation (9), we focused on clearly conveying the core structure of the conic reflow loss. For simplicity, we presented the expectation over time $t$, treating $\epsilon$ and $\zeta$ as fixed samples per iteration.
>
> While it is more mathematically precise to express the full expectation over all random variables $\epsilon$, $\zeta$,and $t$, omitting some expectations is a common simplification in generative modeling literature, especially when the omitted variables are re-sampled at every training step.
>
> That said, we appreciate the reviewer’s suggestion and will denote the sampling of $\epsilon$ and $\zeta$ more explicit in the final version for improved clarity and rigor.
>
> ## W4
>
> We identify a meaningful gap between:
> - **Reconstruction error**:
> $L_{\text{recon}}^2(X) = \mathbb{E}_{x \sim \mathcal{X}} \| x - v(v^{-1}(x)) \|^2$
> - **Perturbed reconstruction error**:
> $L_{\text{p-recon}}^2(X, \varepsilon) = \mathbb{E}_{x \sim \mathcal{X}, z \sim \pi_0} \| x - v(v^{-1}(x) + \varepsilon z) \|^2$
>
> We argue that **Conic ReFlow effectively mitigates drift** (i.e., bias toward fake samples), and provide the following rationale:
>
> 1. Generative models inherently cannot perfectly learn the ideal target distribution.
> 2. Original ReFlow relies only on fake samples, which boosts 1-step quality but leads to degradation in full-step generation.
> 3. A noticeable gap in reconstruction and perturbed reconstruction error is observed between fake and real images.
> 4. This gap indicates that the model is **biased toward fake images** and lacks robustness around real samples.
> 5. Conic ReFlow anchors on the reverse noise of real samples and supervises its perturbed path directly.
> 6. As a result, it narrows the real/fake error gap and alleviates degradation in full-step quality.
>
> **Conic Reflow is theoretically sound and geometrically motivated.**
> Importantly, reconstruction error is a widely adopted proxy for assessing model robustness, sampling stability, and anomaly detection capability across various generative modeling approaches [1–5], and training methods based on input perturbation are commonly used to reduce model bias or improve performance [6–9].
>
> Our method learns not just to map a point but to **cover the cone-shaped neighborhood around real data on the manifold**. This enables training of geometry-aware flows that remain robust in real data regions—something not achievable by straight paths generated from fake-only inversions.
>
> [1] Clustering and Unsupervised Anomaly Detection with L2 Normalized Deep Auto-Encoder Representations
> [2] Anomaly Detection with Robust Deep Autoencoders
> [3] Learning Deep Representations of Appearance and Motion for Anomalous Event Detection
> [4] Šmídl et al., Anomaly Scores for Generative Models (2019)
> [5] Adversarial Examples for Generative Models, ICLR Workshop (2020)
> [6] Input Perturbation Reduces Exposure Bias in Diffusion Models (arXiv:2404.01734)
> [7] LaRE²: Latent Reconstruction Error Based Method for Diffusion-Generated Image Detection
> [8] PixelDP: Leveraging Adversarial Perturbations for Robust Supervised Learning (NeurIPS 2018)\
> [9] SlimFlow: Training Smaller One-Step Diffusion Models with Rectified Flow, ECCV, 2024.
>
> ## W5
>
> 1. The direct performance difference between using only real+fake data ([2]) and applying Slerp can be seen in our ablation study (Section 4.4, Table 2).
> The comparison between** "No-Slerp"** and **"Ours"** clearly demonstrates the **impact of Slerp-based perturbation**.
>
> |Model|FID|IS|Curvature|IVD|Recon (Real)|Recon (Fake)|P-Recon (Real)|P-Recon (Fake)|
> |-|-|-|-|-|-|-|-|-|
> |Ours*|**5.98**|**8.79**|**0.002295**|0.253334|**0.019404**|0.023139|**0.022382**|0.025206|
> |No Slerp|6.60|8.57|0.002322|**0.240884**|0.023380|**0.020154**|0.026063|**0.022496**|
>
> **Strengths of Ours**
>
> 1.Better generation quality.
> 2.Straighter sampling path.
> 3.Lower reconstruction and perturbed reconstruction error on real images.
>
> **Using only real pairs is not sufficient**
> 1. Despite worse generation quality, IVD is lower.
>     - This indicates that the model initially predicts a worse velocity direction compared to ours.
>     - In other words, simply using real pairs does not sufficiently mitigate the bias toward fake samples in standard ReFlow.
>
> 2.**[2] analyzes the **self-consuming training issue** by measuring the magnitude of DAE model weights**.
>
> This is a perspective focused on the training dynamics. In contrast, we target the distributional drift and reconstruction error directly and propose a perturbation-aware conic guidance mechanism. Thus, although both methods aim to improve reflow, our approachs are **trajectory and bias toward fake samples**, and are fundamentally different from [2].
>
> [2] Zhu, H., Wang, F., Ding, T., Qu, Q., & Zhu, Z. (2024). Analyzing and Improving Model Collapse in Rectified Flow Models. arXiv preprint arXiv:2412.08175.
>
> ## Q1
>
> We appreciate the detailed feedback.
>
> - In the final version, we will unify the notation for *k*-rectified flow using **consistent LaTeX math style ($k$)** throughout the text.
>
>
> ## Q2
>
> Thank you for the detailed feedback.
> - The set $\mathbb{N}$ refers to the **total number of training steps** involving both real and fake pairs (Lines 159–160).
> However, we acknowledge that the reference in Line 161 to Section 3.3 for its definition may cause confusion.
> To avoid confusion, we will remove the notation $\mathbb{N}$ from the corresponding line in the final version.
>
> * Equation (11) presents the overall loss function that combines both fake and real pair supervision.
>
> * Equation (12) provides the indexing rule that determines which of the two loss terms is active at each training iteration.
>
> Specifically:
> - If the training index $i \in U_{\text{fake}}$, then the first term $\chi_{\text{fake}} \cdot \left( \dot{Z}_{t,F} - v_\theta(Z_{t,F}) \right)^2$ is activated.
> - If $i \in U_{\text{real}}$, then the second term $\chi_{\text{real}} \cdot \left( X_1 - \text{slerp}(Z_{0,R}, \epsilon, \zeta) - v_\theta(Z_t^{\text{Conic}}) \right)^2$ is used.
>
> Therefore, Equation (12) is not an empirical approximation of (11), but rather describes the index set over which the individual terms in (11) are selected and applied during training.
>
> We hope these clarifications help resolve the reviewer’s concerns We hope this addresses your concern regarding computational efficiency, and we would be glad to provide further clarification during the discussion phase.

---

> > ### Comment · Reviewer_srv3 · 2025-08-01
> >
> > Thank you for your response. Numerical experiments show that using slerp improves the result. Is there any intuitive reason why it is the case? Is there something special about slerp, or are there any other form of interpolation that also enjoys similar properties? In particular, can you elaborate on the following comment that you made:
> >
> > *Our method learns not just to map a point but to cover the cone-shaped neighborhood around real data on the manifold. This enables training of geometry-aware flows that remain robust in real data regions—something not achievable by straight paths generated from fake-only inversions.*

---

> > > ### Author Response · Authors · 2025-08-03
> > > **Additional comment by Authors**
> > >
> > > Thank you for your thoughtful follow-up question, and we’re glad that you agree with the empirical effectiveness of our approach.
> > >
> > > The intuition behind using **Slerp** (spherical linear interpolation) in our Conic ReFlow is rooted in preserving geometric consistency on the data manifold. In particular:
> > >
> > > 1. In high-dimensional spaces, samples from a standard Gaussian tend to lie near the surface of a sphere [1,2].
> > > 2. Conventional Lerp interpolation generates trajectories that do not stay on the surface of the n-dimensional sphere, leading to paths that deviate from the high-density region of the latent distribution [4].
> > > 3. As a result, Lerp may not provide sufficiently reliable supervision for the perturbed paths around the reverse noise of real images.
> > > 4. In contrast, **Slerp** enables smooth interpolation over the sphere and avoids cross-over among multiple perturbed paths—preserving the **conic structure** around real images in single cone (see Fig. 4(b)).
> > > 5. Therefore, our use of Slerp is intentionally designed to supervise straight, geometry-preserving paths emanating from real samples.
> > >
> > > Indeed, in the context of generative models like diffusion, recent works [3,4,5] have shown that **Slerp-based interpolation better preserves geometric structures** and latent semantics than Lerp, further supporting our choice.
> > >
> > > We sincerely appreciate your detailed and valuable question. We hope this explanation helps clarify the reasoning behind our design, and we would be happy to further elaborate on any follow-up questions during the discussion phase.
> > >
> > > References
> > > [1] Gaussian distributions in high dimensions: Concentration of measure, 2021.\
> > > [2] The Concentration of Measure Phenomenon. Number 89 in Mathematical Surveys and Monographs. American Mathematical Society, 2001.\
> > > [3] Image interpolation with scroe-based riemannian metrics of diffusion models ICLR workshop, 2025.\
> > > [4] Unsupervised Discovery of Semantic Latent Directions in Diffusion Models, https://arxiv.org/pdf/2302.12469, 2023.\
> > > [5] Addressing degeneracies in latent interpolation for diffusion models, https://arxiv.org/pdf/2505.07481, 2025.

---

> ### Comment · Reviewer_srv3 · 2025-08-04
>
> Thank you for clarifying this. I am okay if the suggested changes are implemented in the revised version. I have increased my score.

---

### Official Review · Reviewer_ABQ1 · 2025-06-09

**Clarity:** 2
**Significance:** 2
**Originality:** 2
**Rating:** 4
**Confidence:** 5

**Summary:**

This paper addresses ReFlow, which is a procedure for straightening flow trajectories by training flow models upon noise-image pairs generated by itself. While ReFlow does produce faster flow models, the generated marginal drifts away from true marginals as a result of training with fake pairs (pairs generated by solving the flow starting from noise to data). While training with real pairs (pairs generated by solving the flow starting from data to noise) can alleviate this drifting problem, one can only generate a finite number of real pairs.

To mitigate this problem, the authors propose to augment real with with **conic pairs**. Given a real pair, a conic pair is produced by spherically interpolating generated noise with random noise. ReFlow with conic pairs is shown to out-perform vanilla ReFlow on CIFAR10, ImageNet 64x64, and LSUN Bedroom datasets.

**Questions:**

I am willing to raise the score to *borderline reject* if the authors address concerns in

- **Clarity** with further explanations or proofs,
- **Originality** by adding proper baselines,

and further to *borderline accept* if the authors address concerns in

- **Significance** by demonstrating that conic ReFlow can achieve results (e.g., FID scores) comparable to those of recent ReFlow models such as RF++ and Simple ReFlow.

**Ethical Concerns:**

["NO or VERY MINOR ethics concerns only"]

**Final Justification:**

The rebuttal has partially addressed my concerns with

- Significance [W1] with experiments on CIFAR10 (but experiments on ImageNet is missing),
- Clarity [W1], [W2], [W3] with further discussion.

However, some concerns with Originality [W1], [W2] still remain unaddressed:

- No numerical comparisons to relevant baselines such as noise annealing in SlimFlow and real-fake pair balancing in Simple ReFlow,
- Incremental nature of the proposed method compared to previous works.

Hence, I have raised the score to **borderline reject**.

==========

Updated 25.08.12

The authors' latest response has addressed my remaining concerns, but **additional experiments with noise annealing and SlimFlow and real-fake pair balancing in Simple ReFlow should be included in the camera ready version, if accepted**. I tentatively raise my score to **borderline accept**.

**Limitations:**

The authors discuss some limitations in the conclusion.

**Paper Formatting Concerns:**

None.

**Quality:**

3

**Strengths And Weaknesses:**

Strengths and weaknesses are denoted [S#] and [W#], respectively.

**Quality**
- [S1] The paper provides thorough ablations along with experiments on standard benchmark datasets such as CIFAR10, ImageNet, and LSUN Bedroom.

**Significance**
- [S1] Conic ReFlow may be applied to larger flow models such as InstaFlow and SD3.
- [W1] The performance of conic ReFlow is rather weak when compared to recent ReFlow models such as [2] or [3]. For instance, on CIFAR10, while conic ReFlow achieves 3.24 FID with NFE=104, [2] achieves 1.98 FID with NFE=9, and [3] achieves 3.07 FID with NFE=1. On ImageNet 64x64, while conic ReFlow achieves 28.2 FID with full simulation (NFE unreported), [2] achieves 1.74 FID with NFE=9, and [3] achieves 4.31 FID with NFE=1.

**Clarity**
- [S1] The experimental protocol is described clearly.

I had difficulty following the logic at some parts of the paper. See below for details.

- [W1] At lines 104-109, the authors assert that continuity near real samples imply closeness of generative and real marginals. Why should this be the case? The former property is a local one, and the latter property is a global one, and in general, one does not imply the other.
- [W2] At lines 146-148, the authors claim "this approach (conic ReFlow) not only mitigates distributional drift but is also aligned with adversarial robustness strategies studied in inverse problems and reconstruction-based neural networks". Given the ambiguity described in [W1], I have difficulty seeing how conic ReFlow mitigates distributional drift.
- [W3] In Section 4.4.1, the authors propose refreshing real pairs every fixed number of model updates. However, it is unclear why one should do so. Wouldn't it be better just to generate a large number of real pairs prior to training?

**Originality**
- [S1] This paper proposes a new type of augmentation for real pairs for ReFlow training.
- [W1] A similar augmentation has been explored in [1] (albeit with different motivations). Specifically, [1] proposes annealing ReFlow, where, given a noise-image pair, one interpolates noise with random gaussian noise.
- [W2] The idea of balancing training with real and fake pairs has also been suggested in [2].

[1] SlimFlow: Training Smaller One-Step Diffusion Models with Rectified Flow, ECCV, 2024.

[2] Simple ReFlow: Improved Techniques for Fast Flow Models, ICLR, 2025.

[3] Improving the Training of Rectified Flows, NeurIPS, 2024.

---

> ### Author Rebuttal · Authors · 2025-07-31
>
> We appreciate the reviewer for positively noting our ablation studies,  potential applicability to larger flow models.
>
> ## W1
>
> We thank the reviewer for the detailed comparison. However, we respectfully argue that the evaluation that Conic ReFlow underperforms compared to [2] and [3] is not fully justified for the following reasons.
>
> While the results reported in the main paper of [2] are based on a different configuration with batch size 512 and 800K iterations, we apply Conic ReFlow to RF++ under the same setting as the original Rectified Flow, using batch size 128 and 500K iterations, to ensure a fair comparison.
>
> As shown in Table 1 and Table 6 in Appendix G, our method achieves better FID and IS scores under this aligned setting.
>
> * Same configurations
> EDM initialization, LPIPS-Huber-1/tloss, batch size 128 with 500K training itration.
> * Different configurations:
> RF++ / RF++(+Ours) uses 800K/**600K** fake pairs.
>
> |Model|NFE|IS↑|FID↓|
> |-|-|-|-|
> |RF++(+Ours)|1|**9.15**|**3.84**|
> |RF++|1|9.04|4.14|
> |RF++(+Ours)|2|**9.36**|**3.03**|
> |RF++|2|9.24|3.16|
>
> Notably, even with fewer training pairs, our method outperforms the baseline, highlighting that Conic ReFlow consistently improves upon standard reflow even when applied to stronger flow models like [2].
>
> Conic ReFlow is not intended to compete with existing rectified flow techniques but rather to offer a complementary improvement. Our primary objective is to alleviate the drift issue that arises during the reflow process.
>
> In contrast, [2] and [3] focus on overall generation performance improvements, such as EDM initialization, refined loss, and better samplers. Conic ReFlow provides a structural contribution by improving the stability during the reflow.
>
> Therefore, to ensure a fair comparison, all experiments in our paper are conducted using Rectified Flow as the baseline initialization. In addition, the reported results on ImageNet are based on **unconditional generation**, so comparing image quality against class-conditional generation, which benefits from explicit guidance, is not appropriate.
>
> We also clarify that the purpose of **Table 4 and 5** is to empirically demonstrate that our proposed Conic ReFlow consistently improves overall generation quality (FID, IS, and recall) over the original ReFlow step when applied to the same baseline, particularly on more complex datasets. It is **not intended as a direct comparison against current state-of-the-art flow models**.
>
> Our contribution goes beyond merely improving FID scores. The primary goal of Conic ReFlow is to evaluate and enhance the straightness of the reflow path, diversity, fidelity, and mitigation of drift. To this end, we employ a variety of metrics:
>
> - Straightness, IVD
> - Precision / Recall
> - $L_{2}^{recon}, L_{2}^{p-recon}$ and these gaps
>
> Our method demonstrates consistent improvements across diverse datasets, including CIFAR-10, ImageNet 64x64, and LSUN.
>
> In fact, the most methodologically similar approach to ours is SlimFlow [1], which also employs an Annealing ReFlow strategy using noise interpolation for performance gains:
>
> |Model|NFE|FID↓|
> |-|-|-|
> |Slim (EDM)|1|4.53|
> |Slim (1-RF)|1|5.81|
> |Ours (Dist)|1|**4.16**|
> |Ours (RF++)|1|**3.84**|
>
> Conic ReFlow is a novel approach to structurally address the inherent issues of the ReFlow process. This contribution is complementary rather than competitive to prior methods, and its validity is supported by extensive experimental evidence.
>
> [1] SlimFlow: Training Smaller One-Step Diffusion Models with Rectified Flow, ECCV, 2024.\
> [2] Improving the Training of Rectified Flows, NeurIPS, 2024.\
> [3] Simple ReFlow: Improved Techniques for Fast Flow Models, ICLR, 2025.
>
> ## W2
> We thank the reviewer for their detailed and thoughtful questions.
>
> Our claim is that discontinuities near the data manifold can indicate potential global drift. We do not claim that  continuity is theoretically a sufficient condition for global alignment.
>
> The connection between the local continuity and global continuity stems from the structural nature of flow models, where the global vector field is **gradually constructed through the accumulation of conditional path supervision** [1]. Specifically, recent work leverage local continuity as a means to improve global performance, including generalization guarantees under locally Lipschitz conditions [2, 8]. It is a commonly accepted view that local behavior can influence the overall quality of generation.
>
> ## W3
>
> Thank you for the valuable comment. We have quantitatively demonstrated that Conic ReFlow mitigates drift (i.e., bias toward fake images), and this is supported by the following evidence:
>
> 1. Generative models inherently cannot perfectly learn the ideal target distribution.
> 2. Original ReFlow relies only on fake samples, which boosts 1-step quality but leads to degradation in full-step generation.
> 3. A noticeable gap in reconand p-recon error is observed between fake and real images.
> 4. This gap indicates that the model is **biased toward fake images** and lacks robustness around real samples.
> 5. Conic ReFlow anchors on the reverse noise of real samples and supervises its perturbed path directly.
> 6. As a result, the error gap between real and fake samples is reduced, the degradation in full-step quality is mitigated, and our reflow method has been empirically shown to outperform in various metrics such as recall, precision, curvature, and IVD.
>
> Importantly, reconstruction error is a widely adopted proxy for assessing model robustness, sampling stability, and anomaly detection capability across various generative modeling approaches [3–7], and training methods based on input perturbation are commonly used to reduce model bias or improve performance [8–10].
>
> [1] Flow Matching for Generative Modeling\
> [2] De Bortoli et al., Convergence and Generalization of Score-Based Generative Models*(2023)\
> [3] Clustering and Unsupervised Anomaly Detection with L2 Normalized Deep Auto-Encoder Representations, CoRR, 2018\
> [4] Anomaly Detection with Robust Deep Autoencoders, KDD 2017\
> [5] Learning Deep Representations of Appearance and Motion for Anomalous Event Detection, BMVC 2015\
> [6] Šmídl et al., Anomaly Scores for Generative Models, arXiv:1905.11890, 2019\
> [7] Adversarial Examples for Generative Models, ICLR Workshop, 2020\
> [8] Input Perturbation Reduces Exposure Bias in Diffusion Models, ICML, 2023\
> [9] LaRE²: Latent Reconstruction Error Based Method for Diffusion-Generated Image Detection, CVPR, 2024\
> [10] PixelDP: Leveraging Adversarial Perturbations for Robust Supervised Learning (NeurIPS 2018), NeurIPS, 2018\
>
> ## W4
>
> We justify our design choice based on both intuitive considerations and theoretical motivations from recent literature:
>
> 1. Reverse noise are inherently imperfect : These pairs are generated through imperfect inversion processes. If reused without refresh, they can cause the model to overfit or saturate to suboptimal or outdated couplings.
>
> 2. Motivation from Minibatch and Multisample Coupling Strategies : As shown in  [1] and [2], jointly constructed couplings over minibatches or multisamples better approximate ideal transport plans than independent pairings. These methods explicitly construct $\pi(x,y)$ on a batch to reduce training variance and improve flow straightness and transport cost.
>
> 3. Bias mitigation and perturbation-aware stability : Periodic refreshing prevents bias from stale supervision and ensures that reverse noise perturbations act on diverse, up-to-date pairs—promoting stability and robustness during training.
>
> [1] Improving and Generalizing Flow-Based Generative Models with Minibatch Optimal Transport, TMLR , 2024.\
> [2] Multisample Flow Matching: Straightening Flows with Minibatch Couplings, PMLR , 2023.
>
> ## W5
>
> We respectfully clarify that while our method and Slim Flow [1] both involve perturbed supervision, the motivation, design intent, and experimental behavior differ significantly:
>
> Slim Flow introduces Annealing Reflow solely to provide an efficient initialization for a smaller student model by gradually transitioning the target velocity from random to teacher model.
>
> In contrast, our Conic Reflow is designed to improve the reflow process itself, by correcting biases introduced by imperfect real noise couplings. Our method is applicable to models of any size and is not limited to distillation settings.
>
> Slim Flow only evaluates 1-step generation quality, as its objective is to distill a teacher model's immediate behavior. It does not report or ensure performance on multi-step sampling.
>
> In contrast, our method directly improves both *few-step and multi-step generation quality* (see Table 1), making it robust for practical generative tasks.
>
> Slim Flow lacks any data-driven or path-aware adaptation of the annealing parameter $\beta$.
>
> Conversely, our Conic Reflow includes an adaptive $\zeta_{\max}$ formulation (Eq. 13) that adjusts perturbation intensity, providing more principled and effective supervision.
>
> ## W6
>
> While both methods utilize real data points, the motivations and problem setups are fundamentally different.
> Simple Reflow employs forward pairs as a minimal remedy to mitigate error accumulation.
> In contrast, our method aims to address a more foundational issue by leveraging real and generated image pairs.
> Specifically, we mitigate instability and bias in the reflow process, construct adaptive paths, and enhance training stability.
>
> ## Q1
>
> - Clarity :
>   Through responses to W2 and W3, we provided logical explanations, experiments, and supporting evidence.
> - Originality :
>   Through W5 and W6, we clarified the differences between our method and proper baselines, highlighting the originalit of our approach.
> - Significance:
>   In W1, we clarified that our experiments include a direct comparison under the same configuration.
>
> We hope these clarifications help resolve the reviewer’s concerns and would be glad to provide further clarification during the discussion phase.

---

> ### Comment · Area_Chair_KBEs · 2025-08-04
> **Action Required: Author–Reviewer Discussion Closing Soon**
>
> Dear Reviewer,
>
>
>
> This is a gentle reminder that the **Author–Reviewer Discussion** phase ends within just three days (by **August 6**). Please take a moment to read the authors’ rebuttal thoroughly and engage in the discussion. Ideally, every reviewer will respond so the authors know their rebuttal has been seen and considered.
>
>
>
> Thank you for your prompt participation!
>
>
>
> Best regards,
>
>
>
> AC

---

> ### Comment · Reviewer_ABQ1 · 2025-08-05
>
> Thank you for the thorough rebuttal. The rebuttal has partially addressed my concerns with
>
> - Significance [W1] with experiments on CIFAR10 (but experiments on ImageNet is missing),
> - Clarity [W1], [W2], [W3] with further discussion.
>
> However, some concerns with Originality [W1], [W2] still remain unaddressed:
>
> - No numerical comparisons to relevant baselines such as noise annealing in SlimFlow and real-fake pair balancing in Simple ReFlow,
> - Incremental nature of the proposed method compared to previous works.
>
> Hence, I have raised the score to **borderline reject**.

---

> ### Author Response · Authors · 2025-08-06
> **Additional Comment by Authors**
>
> Thank you for your updated comment, and we’re glad that our previous rebuttal has helped improve clarity.
>
> We agree that including comparisons on ImageNet between RF++[1] and RF++(+ours), as well as applying our method to the models of [2]and [3], would further strengthen the persuasiveness of our contributions. However, we would like to respectfully explain that conducting additional from-scratch experiments involving [2] and [3] within the discussion period is **realistically infeasible due to the following reasons**:
>
> - To perform ReFlow, we must generate **at least 1M fake pairs** for CIFAR10 and ImageNet, respectively.
> - For a fair comparison, we would need to train [1], [2] and [3] with our methods from scratch with their full settings:
>   - [1] uses batch size 2048 with 700K iterations on ImageNet64.
>   - [2] uses batch size 512 with 200K iterations on CIFAR10 and batch size 1024 with 500K iterations on ImageNet64.
>   - [3] uses 1.2M iterations total with an unknown batch size in annealing reflow process.
>
> Unfortunately, given our limited GPU resources, this level of training is not realistically possible within the rebuttal period.
>
> As a feasible alternative, we instead focused on clearly articulating the differences in motivation, problem formulation, key contributions, and methodological design between our method and those in [2,3].
>
> Our method tackles the drift phenomenon during ReFlow by introducing **perturbation-based supervision** on conic paths around real samples. This is fundamentally distinct from the noise interpolation of [2] or the fixed real-fake balancing of [1], both in motivation and design.
>
> Empirically, we also note that SlimFlow’s ablation Table 3 reports an RK-step FID of 5.46 with annealing ReFlow (NFE unspecified), while our RK-step FID reaches **3.24**.
> More notably, our **1-step distilled model** achieves **FID 4.16**, surpassing even the RK-solver quality of SlimFlow’s annealing ReFlow baseline.
>
> In addition, our Conic ReFlow offers a general and geometry-aware perspective that may help explain *why* annealing ReFlow empirically works well.
>
>
> While SlimFlow [3] introduces an annealing schedule, the choice of its key hyperparameter $\beta$ is not fully justified. In contrast, our $\zeta_{\text{max}}$ is computed adaptively based on the learned velocity field and the training data distribution, as described in Equation (13). This principled formulation suggests that Conic ReFlow has the potential to generalize better across datasets.
>
> We sincerely hope that our response has clarified the originality and focused contribution of our method. We greatly value your suggestion, and to further address your concern, we plan to include additional empirical comparisons in the **camera-ready version**, where we will apply our Conic ReFlow to [2] and [3] and report generation quality for stronger support.
>
> Once again, thank you for acknowledging the improvement in clarity. We hope that this response further helps address the concerns regarding originality, and we’re happy to provide any further clarifications during the discussion phase.
>
>
>
> [1] Improving the Training of Rectified Flows, NeurIPS, 2024.
>
> [2] Simple ReFlow: Improved Techniques for Fast Flow Models, ICLR, 2025.
>
> [3] SlimFlow: Training Smaller One-Step Diffusion Models with Rectified Flow, ECCV, 2024.

---

### Official Review · Reviewer_MDqu · 2025-07-01

**Clarity:** 3
**Significance:** 2
**Originality:** 3
**Rating:** 4
**Confidence:** 4

**Summary:**

This article reveals that the reflow process causes the learned model to drift away from the target distribution and designs a new reflow step, namely the conic reflow, to address this drift problem. It supervises the model by the inversions of real data points through the previously learned model and its interpolation with random initial points. Conic reflow achieves much lower FID in both one-step and full-step generation in CIFAR-10. The conic reflow generalizes to various datasets such as LSUN Bedroom and ImageNet.

**Questions:**

1、Please refer to the question in the weakness section.

2. The ablation study of selecting slerp or conic hyperparameters is not sufficient. The metric differences between different hyperparameter combinations are too small.

3. I wonder if the conic reflow costs more compute resources than the original Rectified Flow?

**Ethical Concerns:**

["NO or VERY MINOR ethics concerns only"]

**Final Justification:**

I have reviewed the author's rebuttals to all the reviewers, and they have addressed my concerns.

**Limitations:**

Yes.

**Paper Formatting Concerns:**

No.

**Quality:**

2

**Strengths And Weaknesses:**

# Strengths

The article is well written and organized. The article proposes a new type of reflow, named conic reflow. It keeps the ODE paths aligned with real samples, as evaluated by reconstruction. It uses only a small number of generated samples, 600K and 4M, respectively, instead of the large ones. The learned flow is straighter than others, as evaluated by curvature. Experiments demonstrate that learned model generates images with higher quality, evaluated by FID, IS, and Recall.



# Weaknesses

1. The proposed method lacks a thorough theoretical analysis of the conic flow and its potential to alleviate drift issues. Additionally, the paper does not include a comprehensive analysis for selecting slerp or conic hyperparameters theoretically or empirically.

2. The paper compares performance only with the original rectified flow and does not include a comparison with recent flow matching models or diffusion models.

3. The empirical improvements for the conic reflow in CIFAR-10 are too small, and it does not adequately verify the effectiveness of the conic reflow. The author should include more empirical results (such as higher-resolution ImageNet, FFHQ, AFHQ, or other classes of LSUN).

Based on the above content, I recommend that the author submit additional experimental results for review. If you can provide additional theoretical analysis for the conic reflow, it will also help change the perceptions of the paper.

---

> ### Author Rebuttal · Authors · 2025-07-31
>
> We thank the reviewer for their careful reading and appreciation of our key ideas, including the conic reflow design, drift mitigation, and strong empirical performance across datasets.
>
> ## W1
>
> Thank you for the valuable comment.
>
> While we do not provide a formal theoretical analysis, **Conic ReFlow is logically designed to mitigate drift** (i.e, biased toward fake images). Its effectiveness is supported by empirical results, and its rationale can be reasonably grounded in existing literature on reconstruction-based robustness and reduces Bias.
>
> 1. Generative models inherently cannot perfectly learn the ideal target distribution.
> 2. Original ReFlow relies only on fake samples, which boosts 1-step quality but leads to degradation in full-step generation.
> 3. A noticeable gap in reconand p-recon error is observed between fake and real images.
> 4. This gap indicates that the model is **biased toward fake images** and lacks robustness around real samples.
> 5. Conic ReFlow anchors on the reverse noise of real samples and supervises its perturbed path directly.
> 6. As a result, the error gap between real and fake samples is reduced, the degradation in full-step quality is mitigated, and our reflow method has been empirically shown to outperform in various metrics such as recall, precision, curvature, and IVD.
>
> Importantly, reconstruction error is a widely adopted proxy for assessing model robustness, sampling stability, and anomaly detection capability across various generative modeling approaches [1–5], and training methods based on input perturbation are commonly used to reduce model bias or improve performance [6–9].
>
>
> [1] Clustering and Unsupervised Anomaly Detection with L2 Normalized Deep Auto-Encoder Representations, CoRR, abs/1802.00187, 2018
> [2] Anomaly Detection with Robust Deep Autoencoders, KDD 2017
> [3] Learning Deep Representations of Appearance and Motion for Anomalous Event Detection, BMVC 2015
> [4] Anomaly Scores for Generative Model, arXiv:1905.11890, 2019
> [5] Adversarial Examples for Generative Models, ICLR Workshop, 2020
> [6] Input Perturbation Reduces Exposure Bias in Diffusion Models ICML, 2023
> [7] LaRE²: Latent Reconstruction Error Based Method for Diffusion-Generated Image Detection, CVPR, 2024
> [8] PixelDP: Leveraging Adversarial Perturbations for Robust Supervised Learning, NeurIPS, 2018\
> [9] SlimFlow: Training Smaller One-Step Diffusion Models with Rectified Flow, ECCV, 2024
>
> ## W2
>
> Thank you for the detailed feedback.
>
> Our paper includes comparisons with recent diffusion-based models in Appendix M to further contextualize the performance of reflow-based methods. It is mentioned in L210 and Section 4.6.
>
> We suggest that including comparisons with more recently developed flow-based generative models would help readers better understand the overall generative quality landscape. We will provide FID and IS scores of recent flow models, including [1–4], in Appendix M of the final version.
>
> [1] Improving and Generalizing Flow-Based Generative Models with Minibatch Optimal Transport, TMLR , 2024.\
> [2] Zhu, H., Wang, F., Ding, T., Qu, Q., & Zhu, Z. (2024). Analyzing and Improving Model Collapse in Rectified Flow Models. arXiv preprint arXiv:2412.08175.\
> [3] SlimFlow: Training Smaller One-Step Diffusion Models with Rectified Flow, ECCV, 2024.\
> [4] Simple ReFlow: Improved Techniques for Fast Flow Models, ICLR, 2025.\
>
> ## W3
>
> We appreciate the reviewer’s request for additional high-resolution experiments.
>
> We would like to note that we have already included high-resolution results on the LSUN Bedroom dataset in our current submission, covering FID, precision, recall, $L_2^{recon}$, and $L_2^{p\text{-}recon}$ (see Figure 8). We also provide qualitative comparisons across random seeds in Appendix (Figures A6–A10), demonstrating the robustness and effectiveness of Conic ReFlow.
>
> However, further experiments on other high-resolution datasets such as LSUN-Church, AFHQ, or ImageNet 256 are realistically infeasible within the rebuttal period due to the following constraints:
>
> 1. The original Rectified Flow repository only provides 1-Rectified Flow checkpoints for these datasets, requiring us to re-train 2-Rectified Flow baselines from scratch for fair comparison.
>
> 2. High-resolution datasets of this scale require at least 120K fake samples to produce meaningful comparisons under the reflow paradigm.
>
> 3. Given our limited GPU resources, training with batch size 16 would take **approximately 4 weeks** to reach convergence and obtain statistically meaningful results.
>
> We sincerely regret that we are unable to include these additional results in the rebuttal. However,** we plan to report full evaluations on LSUN-Church and AFHQ datasets in the camera-ready version**, including comparison with the original Rectified Flow and release of corresponding model checkpoints.
>
> We hope this addresses your concerns and clarifies the practical limitations involved.
>
> ## Q2
>
> We appreciate the reviewer’s concern regarding hyperparameter sensitivity.
>
> ### We provide several ablation studies on different Slerp settings.
> For **Slerp perturbation patterns**, we provide detailed comparison in **Table 3**, showing that the interpolation method yields consistent quality across different variants.
> - Ours Slerp Pattern: $\zeta_t=\zeta_{\text{max}}/K \rightarrow\zeta_{\text{max}} \rightarrow \zeta_{\text{max}}/K$
> - Strictly Increasing:$\zeta_{t}=\zeta_{\text{max}}/K \rightarrow\zeta_{\text{max}}$
> - Strictly Decreasing: $\zeta_{t}=\zeta_{\text{max}}\rightarrow\zeta_{\text{max}}/K$
>
> |Category|Method|IS (↑)|FID (↓)|
> |-|-|-|-|
> |Slerp|Ours|**8.72**|**6.63**|
> ||Strictly Increasing|8.48|6.64|
> ||Strictly Decreasing|8.45|6.70|
> |Lerp| Linear Interpolation |8.46|7.50|
>
> Futhermore, appendix E Table A4 compares the IS/FID performance when using only real pairs, using both real and fake pairs, and applying our conic reflow method.
>   - Real-only pairs (Original Rectified flow using only Real pair),
>   - Real + Slerp-perturbed pairs,
>   - Real + Fake + Slerp pairs
>
> |Setting|1-step IS|1-step FID|RK-step IS|RK-step FID|
> |-|-|-|-|-|
> |Real only (no Slerp)| 8.21|6.93|8.65|5.13|
> |Real only (+ Slerp)| 8.53|6.71|9.02|4.47|
> |Ours (real + fake + Slerp)|**8.79**|**4.16**|**9.30**|**3.24**|
>
> Overall, the results empirically support that conic reflow consistently outperforms standard reflow regardless of the presence of fake pairs or the specific noise injection pattern used.
>
> ### 2. $\zeta_{max}$ is adaptively selected
>
> We would like to clarify that:
> $\zeta_{max}$ is not manually tuned but **adaptively selected** based on the maximum p-reconstruction error between fake and real samples (eqn 13 in Sec3.5).
>
> We acknowledge that Table 3 may have caused confusion by presenting a fixed value for $\zeta_{\max}$. In the final version, we will revise this to clarify the scheduling pattern:
> - For strictly increasing : $\zeta_{t}=\zeta_{\text{max}}/K \rightarrow\zeta_{\text{max}}$
> - For strictly decreasing : $\zeta_{t}=\zeta_{\text{max}}\rightarrow\zeta_{\text{max}}/K$
>
> Once again, thank you for the helpful suggestion, and we hope this response has clarified your concerns.
>
> ## Q3
> Thank you for the thoughtful question.
>
> We would like to clarify that the computational cost of conic reflow is essentially the same as that of the original rectified flow, except for the initial step of estimating $\zeta_{\max}$.
>
> Importantly, conic reflow is significantly more efficient in terms of total training resources. Unlike the baseline our method achieves strong performance using only about **7–10% of fake pairs**. As a result, the overall training cost is actually **lower** than that of the baseline.
>
> We hope this addresses your concern regarding computational efficiency, and we would be glad to provide further clarification during the discussion phase.

---

> > ### Comment · Reviewer_MDqu · 2025-08-05
> > **Response to Author**
> >
> > Thank you for your response. I have reviewed the author's rebuttals to all the reviewers, and they have addressed my concerns. I look forward to seeing the changes from the rebuttals reflected in the revised version. I have increased my score.

---

> ### Comment · Area_Chair_KBEs · 2025-08-04
> **Action Required: Author–Reviewer Discussion Closing Soon**
>
> Dear Reviewer,
>
>
>
> This is a gentle reminder that the **Author–Reviewer Discussion** phase ends within just three days (by **August 6**). Please take a moment to read the authors’ rebuttal thoroughly and engage in the discussion. Ideally, every reviewer will respond so the authors know their rebuttal has been seen and considered.
>
>
>
> Thank you for your prompt participation!
>
>
>
> Best regards,
>
>
>
> AC

---

### Official Review · Reviewer_AnKE · 2025-07-03

**Clarity:** 3
**Significance:** 2
**Originality:** 2
**Rating:** 4
**Confidence:** 4

**Summary:**

This paper first demonstrates that the original reflow process in Rectified Flow relies on large amounts of generated data and can cause drift from the target distribution. To address this, the authors propose Balanced Conic Rectified Flow, which incorporates real image inversions and spherical linear interpolation. This approach achieves better performance while requiring significantly fewer generated samples.

**Questions:**

See Weakness. Also,
- The arrangement and legends in Figures 3a–b and 5b could be improved for better readability.
- The first footnote on page 3 is somewhat confusing. It refers to "the t-th sampling step", and adds the velocity at time $t$ directly to $X_0$ or $X_1$, which does not make sense. It seems that the process being described should involve integration from $t = 0$ to $t = 1$, or vice versa, rather than referring to a single "t-th sampling step".

**Ethical Concerns:**

["NO or VERY MINOR ethics concerns only"]

**Final Justification:**

The authors have adequately addressed my concerns regarding [W1] the use of 1-step Euler sampling and [W2] the stability of the hyperparameter settings. I appreciate the clarifications and will maintain my current borderline accept score.

**Limitations:**

yes

**Quality:**

3

**Strengths And Weaknesses:**

Strengths:
- The proposed method outperforms reflow and rf++, with experiments on diverse datasets including CIFAR-10, ImageNet, and LSUN.
- The drift analysis is insightful and serves as a good motivation.

Weaknesses:
- The paper analyzes drift using 1-step Euler sampling; however, Rectified Flow models are not designed for 1-step generation, and fake samples for the reflow process are generated via multi-step integration. This raises concerns that the measured drift may reflect Euler integration error rather than actual model drift, potentially biasing the conclusions. Clarification or validation with a multi-step analysis would strengthen the argument.
- The proposed method introduces several hyperparameters, including the maximum Slerp interpolation weight $\zeta^{max}$, the repairing step, and the mixing ratio between real and fake pairs in balanced conic reflow. However, the paper lacks sufficient analysis of how sensitive the model's performance is to these hyperparameters. It is unclear whether the reported improvements are robust or heavily dependent on careful tuning.
- While the paper empirically demonstrates that using Slerp-based perturbations around real image inversions helps mitigate drift, it does not offer a theoretical explanation for why this technique effectively addresses the issue.

---

> ### Author Rebuttal · Authors · 2025-07-31
>
> We appreciate the reviewer’s agreement with our motivation and contributions, especially the drift analysis and improvements over existing reflow methods.
>
> ## W1
> We appreciate the reviewer’s concern and the opportunity to clarify our choice of analysis. We suggest that 1-step reconstruction is an appropriate and informative diagnostic for drift (i.e., bias), for the following reasons:
>
> 1. **Support from prior work**: Prior studies on exposure bias in diffusion models (e.g., [1], [2]) emphasize that the early steps of sampling are where the most critical errors or biases arise. Our use of 1-step evaluation is consistent with this literature.
>
> 2. **Reduced solver accumulation error**: Multi-step sampling introduces numerical errors that accumulate with each step, potentially masking model-induced error. 1- or 2-step evaluation avoids this confounding factor and more cleanly isolates the model’s contribution to reconstruction performance.
>
> 3. **Direct signal of drift** (bias toward fake samples): Since the reflow trajectory heavily depends on the model’s predicted initial velocity, a 1-step Euler reconstruction error directly reflects the alignment of the model’s estimated vector field with the true reverse direction.
>
> 4. **Statistical robustness**: We report reconstruction errors averaged over repeated evaluations on 10k samples using the same solver settings. The consistent error gap between real and fake samples likely reflects intrinsic model behavior, not solver stochasticity.
>
> 5. **Multi-step support in supplement**: In Supplementary Figure 5, we also report 2-step Euler reconstruction error on ImageNet. The gap between fake and real reconstruction errors remains, and our proposed conic method demonstrably narrows this gap.
>
> [1] Elucidating the Exposure Bias of Diffusion Models ICLR, 2024\
> [2] Input Perturbation Reduces Exposure Bias in Diffusion Models ICML, 2023
>
> ## W2
> We appreciate the reviewer’s concern regarding hyperparameter sensitivity.
> ### 1.We provide several ablation studies on different Slerp settings.
> Regarding the mixing between real and fake pairs, we provide an ablation study covering:
>   - Real-only pairs (Original Rectified flow using only Real pair),
>   - Real + Slerp-perturbed pairs,
>   - Real + Fake + Slerp pairs,
>
> |Setting|1-step IS|1-step FID|RK-step IS|RK-step FID|
> |-|-|-|-|-|
> |Real only (no Slerp)| 8.21|6.93|8.65|5.13|
> |Real only (+ Slerp)| 8.53|6.71|9.02|4.47|
> |Ours (real + fake + Slerp)|**8.79**|**4.16**|**9.30**|**3.24**|
>
> demonstrating that performance improves consistently when our full method is applied (Supp. Fig. A4).
>
>
> For **Slerp perturbation patterns**, we provide detailed comparison in **Table 3**, showing that the interpolation method yields consistent quality across different variants.
> - Ours Slerp Pattern: $\zeta_t=\zeta_{\text{max}}/K \rightarrow\zeta_{\text{max}} \rightarrow \zeta_{\text{max}}/K$
> - Strictly Increasing:$\zeta_{t}=\zeta_{\text{max}}/K \rightarrow\zeta_{\text{max}}$
> - Strictly Decreasing: $\zeta_{t}=\zeta_{\text{max}}\rightarrow\zeta_{\text{max}}/K$
>
> |Category|Method|IS ↑|FID ↓|
> |-|-|-|-|
> |Slerp|Ours|**8.72**|**6.63**|
> ||Strictly Increasing|8.48|6.64|
> ||Strictly Decreasing|8.45|6.70|
> |Lerp| Linear Interpolation |8.46|7.50|
>
>
> ### 2. $\zeta_{max}$ is adaptively selected
>
> We would like to clarify that:
>
> $\zeta_{max}$ is not manually tuned but **adaptively selected** based on the **maximum p-reconstruction error** between fake and real samples (eqn 13 in Sec3.5).
>
> To further address the reviewer’s concern, we will additionally report generation quality across varying fake-to-real pair ratios, comparing both the baseline and our method.
>
> |#Fake Pairs|NFE|IS↑|FID↓|
> |-|-|-|-|
> |Original 4M (+Distill)|1|8.08 (9.01)|12.21 (4.85)||
> ||110|9.24|3.36|
> |Ours 600K(+Distill)|1|**8.79 (9.11)**|**5.98 (4.16)**|
> ||104|**9.30**|**3.24**|
> |Ours 3M(+Distill)|1|8.69 (8.92)|6.03 (4.72)|
> ||104|9.20|3.35|
> |Ours 60K|1|8.60|7.06|
> ||104|9.17|3.98|
>
> **Key takeaways:**
> 1. Conic ReFlow improves 1-step performance across all settings.
> 2. 600K achieves the best trade-off between efficiency and performance.
> 3. Too few fake pairs (e.g., 60K) results in poor full-step generalization.(For detailed analysis, see Appendix E.)
>
> These results collectively indicate that our approach maintains robust performance across a range of hyperparameter configurations, and the reported improvements are not heavily reliant on specific tuning choices.
>
> We hope this alleviates the reviewer’s concern.
>
> ## W3
>
> Thank you for the valuable comment.
>
> We argue that **Conic ReFlow effectively mitigates drift** (i.e., bias toward fake samples), and provide the following rationale:
>
> 1. Generative models inherently cannot perfectly learn the ideal target distribution.
> 2. Original ReFlow relies only on fake samples, which boosts 1-step quality but leads to degradation in full-step generation.
> 3. A noticeable gap in recon and p-recon error is observed between fake and real images.
> 4. This gap indicates that the model is **biased toward fake images** and lacks robustness around real samples.
> 5. Conic ReFlow anchors on the reverse noise of real samples and supervises its perturbed path directly.
> 6. As a result, the error gap between real and fake samples is reduced, the degradation in full-step quality is mitigated, and our reflow method has been empirically shown to outperform in various metrics such as recall, precision, curvature, and IVD.
>
> Importantly, reconstruction error is a widely adopted proxy for assessing model robustness, sampling stability, and anomaly detection capability across various generative modeling approaches [1–5], and training methods based on input perturbation are commonly used to reduce model bias or improve performance [6–9].
>
>
> [1] Clustering and Unsupervised Anomaly Detection with L2 Normalized Deep Auto-Encoder Representations
> [2] Anomaly Detection with Robust Deep Autoencoders
> [3] Learning Deep Representations of Appearance and Motion for Anomalous Event Detection
> [4] Šmídl et al., Anomaly Scores for Generative Models (2019)
> [5] Adversarial Examples for Generative Models, ICLR Workshop (2020)
> [6] Input Perturbation Reduces Exposure Bias in Diffusion Models, ICML, 2023
> [7] LaRE²: Latent Reconstruction Error Based Method for Diffusion-Generated Image Detection
> [8] PixelDP: Leveraging Adversarial Perturbations for Robust Supervised Learning, NeurIPS, 2018
> [9] SlimFlow: Training Smaller One-Step Diffusion Models with Rectified Flow, ECCV, 2024.
>
> ## Q1
> Thank you for the helpful suggestion.
> We acknowledge that the readability of Figures 3a–b and 5b can be improved and will revise the arrangement and legends in the final version as follows to enhance clarity:
>
> - For **Figure 3(a)**, we will move the legend to the top center and increase the font size of both the legend and the values above the bars.
> - For **Figure 3(b)**, we will remove the duplicated legend and relocate the remaining one to the top right, increasing the font size to improve readability.
>
> Appreciate your attention to detail.
>
> ## Q2
> Thank you for the helpful suggestion. We agree that the current description is misleading, as it omits the integration process and may imply a direct addition of velocity at a single time step.
> We will add that integration over time is required in the final version.
>
> We would be happy to elaborate over the discussion phase

---

> > ### Comment · Reviewer_AnKE · 2025-08-05
> >
> > Thank you for addressing my concerns regarding [W1] the use of 1-step Euler sampling and [W2] the stability of the hyperparameter settings. With respect to [W3], my request was for a theoretical explanation rather than empirical observations or intuition. I appreciate your clarifications and will keep my current borderline accept score.

---

> ### Author Response · Authors · 2025-08-07
> **Additional Comment by Authors**
>
> We're glad to see that [W1] and [W2] concerns have been addressed, and hope the following theoretical clarification helps resolve your question regarding our perturbation-based supervision.
>
> ---
>
> **Fundamental Limitation of Ideal Target distribution Analysis**
> As the true data distribution is generally unknown in practice, mathematically deriving how close our supervision lies to an ideal target distribution remains a largely open problem. To the best of our knowledge, there exists no comprehensive framework for characterizing this distance in a nonparametric setting.
>
> ---
>
> **Theoretical Foundations of Local-to-Global Flow Construction with Perturbed Path Supervision**
> Flow-based models construct the global vector field gradually through the accumulation of local supervision signals along individual paths [1]. This motivates our use of perturbation-based supervision near real data.
>
> Importantly, recent theoretical developments [2] provide rigorous support for the stability of such localized training when the data distribution $p^\star \in \mathcal{A}_K^\beta$ satisfies mild regularity assumptions (e.g., $\beta$-Hölder smoothness, subGaussian tails).
>
> In particular:
>
> - **Proposition 6** guarantees that the score function $s^\star(t, x)$ is uniformly bounded in space and time.
> - **Proposition 7** establishes one-sided Lipschitz continuity via an exponentially decaying bound on the Jacobian eigenvalues.
> - **Theorem 8** shows that the score is Hölder smooth of arbitrary order on high-probability subsets, with time-dependent norm control.
>
> These results imply that perturbation paths around real samples—such as those used in our method—lie within well-behaved regions of the score field, allowing stable and informative supervision. As a result, our perturbation-based supervision contributes valid gradient signals for constructing the global vector field, without violating the theoretical assumptions required for convergence.
>
> This perspective assumes that the reverse noise associated with real images also lies on a similarly well-behaved manifold—i.e., it is β-Hölder smooth with subGaussian tails, and its score function is bounded and Lipschitz, as in [2]. Under this view, our perturbation-based supervision operates within regular regions of the score field and aligns well with the theoretical foundations of score-based generative modeling.
>
> ---
>
> **Compatibility with Flow Matching Theory**
> Since diffusion models can be interpreted as special cases of flow models under particular path constraints [3,4], the theoretical results from the flow matching literature are directly relevant to our framework. The robustness of perturbed supervision paths is thus supported by the same regularity arguments used to justify flow matching convergence.
>
> ---
>
> **Formulation within Conditional Flow Matching (CFM) Framework**
> Our perturbation-based approach adheres fully to the conditional flow matching (CFM) formulation proposed in [1]. Specifically, our training objective can be written as:
>
>
> $\mathcal{L_{CFM}}(\theta) = \mathbb{E}_{t \sim \mathcal{U}[0,1],\ x_1 \sim q(x_1),\ \epsilon \sim \mathcal{N}(0,I)} \left[ \left\| v_t(\hat{x}) - u_t( \hat{x} \mid x_1) \right\|^2 \right]$
>
> where $\hat{x} = slerp(x,\epsilon, \zeta)$ , and consider conditional probability path of the form :
>
> $p_{t}(\hat{x} \mid x_{1}) = \mathcal{N}(\hat{x} \mid \mu_{t}(x_{1}), \sigma_{t}(x_{1})^2I)$
>
> where $\mu$ and $\sigma$ are time-dependent mean and scalar standard deviation respectively. Given the assumptions above, the theory in [1] guarantees that such loss formulation leads to a well-posed and convergent estimation of the global vector field.
>
> ---
>
> In summary, while we do not provide a formal proof that our perturbation-based supervision approximates an ideal target distribution more closely (nor do we claim a new theoretical bound in this work), we do provide theoretical grounding that:
>
> - Our approach operates within the well-established flow matching framework,
> - It inherits its convergence and regularity properties under mild and reasonable assumptions,
> - And it introduces no violations to the theoretical assumptions required for flow training stability and robustness.
>
> We sincerely hope this explanation alleviates your concerns regarding the theoretical basis of our method. Exploring explicit bounds or optimality conditions for perturbed supervision remains an exciting direction for future work, beyond the current scope of this paper.
>
> We would be happy to further elaborate on any follow-up questions during the discussion phase.
>
> [1] Flow Matching for Generative Modeling, ICLR, 2023.
>
> [2] Convergence and Generalization of Score-Based Generative Models, https://arxiv.org/abs/2507.04794, 2023.
>
> [3] Flow Straight and Fast: Learning to Generate and Transfer Data with Rectified Flow, ICLR, 2023.
>
> [4] Rectified Diffusion: Straightness Is Not Your Need in Rectified Flow, ICLR, 2025.

---

> > ### Comment · Reviewer_AnKE · 2025-08-08
> >
> > Thank you for your response to [W3] again. I appreciate the theoretical grounding you provided, and I understand your point that it is inherently difficult to rigorously quantify how closely your perturbation-based supervision aligns with an ideal target. Based on your contribution, I still decide to keep my score at borderline accept. Thank you again for your thoughtful reply.

---

### Comment · Area_Chair_KBEs · 2025-08-01
**Kindly Engage with Author Responses**

Dear Reviewers,





The authors have submitted their responses to your reviews. At your earliest convenience, please take a moment to engage with their replies. Your continued discussion and clarifications will be invaluable in ensuring a fair and constructive review process for all parties.

Thank you again for your thoughtful contributions and dedication.





Warm regards,



Your AC

---

### Note · Authors · 2025-08-15

We sincerely thank the reviewers for their thoughtful feedback, constructive engagement, and for recognizing our strengths in the originality of our problem formulation, the soundness of our methodology, the improvements achieved by our approach, the theoretical validity of our design, and its general applicability. We also thank the AC for their careful oversight of the review process.

We are pleased that this Final Remarks opportunity allows us to further highlight and clarify the contributions and originality of our work to the reviewers, and to communicate these points clearly to the AC.

1. **Addressing distribution shift in standard ReFlow**\
We showed that standard ReFlow suffers from distribution drift, evidenced by consistent gaps in reconstruction (recon) and perturbed reconstruction (p-recon) errors between real and fake images. Conic ReFlow mitigates this by anchoring supervision on perturbed reverse-noise paths of real samples, reducing the recon gap, preserving the target distribution, and producing straighter ODE trajectories. Gains are confirmed across recon, p-recon, precision, recall, curvature, IVD, FID, and IS.

2. **Originality**\
Unlike prior works using DAE weight analysis, random noise annealing, or simple real data injection, Conic ReFlow supervises both real targets and their surrounding reverse-noise paths, offering a new, geometry-aware approach to drift mitigation in ReFlow-based models.

3. **Lower training cost**\
Our method achieves better quality with only a small number of fake pairs compared to standard ReFlow, lowering computational cost while improving multiple metrics.

4. **Robustness to hyperparameters**\
Ablations show consistent superiority to standard ReFlow regardless of Slerp patterns, absence of Slerp, real-only settings, or varying fake/real ratios.

5. **Dataset generality**\
With adaptively chosen maximum Slerp noise, improvements hold across CIFAR-10, ImageNet, and LSUN Bedroom.

6. **Complementarity**\
Applying Conic ReFlow to RF++ under identical settings yields better generation quality, demonstrating compatibility with stronger baselines.

**Summary**\
Conic ReFlow offers an empirically validated and computationally efficient solution to a core limitation of ReFlow. We thank the reviewers for acknowledging our contributions and valuable suggestions, and we look forward to strengthening this work in the camera-ready version.

---

### Decision · Program_Chairs · 2025-09-17

**Decision:**

Accept (poster)

**Comment:**

This work shows that the reflow process can cause the learned model to deviate from the true target distribution and proposes Conic Reflow as a mitigation. Although Conic Reflow introduces additional hyperparameters, rebuttal evidence (in response to Reviewers AnKE and MDqu) suggests limited sensitivity to them. It is encouraged to include these results, along with the additional experiments suggested by Reviewer ABQ1 (e.g., applying the method to Simple ReFlow and SlimFlow), in the revision. Overall, I recommend **acceptance**.